# Transcriptional changes in the mammary gland during lactation revealed by single cell sequencing of cells from human milk

Alecia-Jane Twigger [1,2,3 ✉], Lisa K. Engelbrecht[3], Karsten Bach[1,2], Isabel Schultz-Pernice [3], Sara Pensa[1,2], Jack Stenning[1,2], Stefania Petricca[3,4], Christina H. Scheel[3,5,6 ✉] & Walid T. Khaled [1,2,6 ✉]

Under normal conditions, the most significant expansion and differentiation of the adult mammary gland occurs in response to systemic reproductive hormones during pregnancy and lactation to enable milk synthesis and secretion to sustain the offspring. However, human mammary tissue remodelling that takes place during pregnancy and lactation remains poorly understood due to the challenge of acquiring samples. We report here single-cell transcriptomic analysis of 110,744 viable breast cells isolated from human milk or non-lactating breast tissue, isolated from nine and seven donors, respectively. We found that human milk largely contains epithelial cells belonging to the luminal lineage and a repertoire of immune cells. Further transcriptomic analysis of the milk cells identified two distinct secretory cell types that shared similarities with luminal progenitors, but no populations comparable to hormone-responsive cells. Taken together, our data offers a reference map and a window into the cellular dynamics that occur during human lactation and may provide further insights on the interplay between pregnancy, lactation and breast cancer.

[1] Department of Pharmacology, University of Cambridge, Cambridge, England. [2] Wellcome-MRC Cambridge Stem Cell Institute, Cambridge, England. [3] Institute of Stem Cell Research, Helmholtz Zentrum München, Munich, Germany. [4] Biomedical Center (BMC), Division of Physiological Genomics, Faculty of Medicine, LMU Munich, Munich, Germany. [5] Department of Dermatology, Ruhr-University Bochum, Bochum, Germany. [6] These authors jointly supervised this work: Christina H. Scheel, Walid T. Khaled. ✉email: ajt215@cam.ac.uk; christina.scheel@klinikum-bochum.de; wtk22@cam.ac.uk

The mammary gland undergoes cycles of tissue remodelling throughout a woman's reproductive lifespan that are particularly pronounced during pregnancy, lactation and the return of the mammary gland to its resting state post involution. Determining the differentiation dynamics driving these developmental stages is not only essential to understanding normal mammary gland function but also the origins of breast cancer. The mammary gland consists of a bilayered ductal tree with an inner layer of luminal cells (LCs) that mature into secretory cells during lactation, and a basal network of contractile myoepithelial (MY) cells that support the transport of milk to the nipple during lactation. In the human mammary gland, this ductal tree is embedded in collagen-rich, specialized stroma containing different types of mesenchymal and immune (IM) cells[1]. Major changes in the architecture and cellular composition of the adult mammary gland are required for the synthesis and secretion of the complex bioactive fluid that is human milk[2]. Findings in murine models suggest that these changes have a lasting impact on the mammary epithelium at an epigenetic level[3] and lead to the generation of parity-induced cell types[4,5]. These molecular and cellular changes occurring in the mammary gland may point toward a mechanism explaining the reduced long-term breast cancer risk associated with parity[6] and extended periods of lactation[7].

Single-cell transcriptomic profiling of murine mammary epithelial cells has shed light on the differentiation dynamics of mammary epithelial cells. These studies described luminal progenitor (LP) cells in the virgin gland that gives rise to hormone-responsive (HR) mature LCs and, in the case of pregnancy, to secretory alveolar cells[5,8,9]. Interestingly, findings from Bach et al. determined that the post-parous mammary gland contained primed parity-induced LP cells that upregulated lactation-associated genes[5]. In addition, recent findings have shown that ageing impacts the composition of the murine mammary gland with the rise of an age-dependent LC subpopulation co-expressing hormone-sensing and secretory-alveolar lineage markers[10]. These findings are of particular interest given that LP cells have been proposed as the cell of origin for different breast cancer subtypes[11]. Analogous to the murine mammary gland, an emerging number of studies have begun to characterize human mammary subpopulations using single-cell transcriptomics[12–16]. Normal mammary tissue is usually derived from aesthetic breast reductions from non-lactating women. Thus, compared to its resting state, tissue from lactating human mammary glands is difficult to obtain.

We report here a single-cell transcriptomic analysis of 110,744 viable breast cells isolated from human milk or non-lactating breast tissue, isolated from 9 and 7 donors, respectively. Our data set that comprised of 56,030 lactation-derived mammary cells (LMCs) or 54,714 non-lactation-derived mammary cells (NMCs) enable us to analyse and compare breast cells between these two states. We found that human milk largely contains epithelial cells belonging to the luminal lineage as well as a sophisticated repertoire of IM cells. Further transcriptomic analysis of the milk cells identified two distinct secretory cell types that share similarities with hormone receptor-negative LPs. Taken together, our data offer a reference map and a window into the cellular dynamics that occur during human lactation and may provide further insights into the interplay between pregnancy, lactation and breast cancer.

## Results

### Cells isolated from milk or breast tissue display distinct molecular profiles. To isolate NMCs, tissue donated from elective aesthetic mammoplasty surgery was mechanically dissected and

enzymatically digested to separate epithelial fragments that could either be immediately frozen or trypsinised further to generate single cells (Fig. 1a and Supplementary Fig. 1a). In contrast, centrifugation of freshly expressed whole milk was sufficient to isolate single LMCs from the pellet of the colloidal suspension (Fig. 1a and Supplementary Fig. 1a). Both viably isolated NMCs and LMCs could be cultured across a range of donors in either two-dimensional (2D) culture plates to generate monolayer cultures (Fig. 1b and Supplementary Fig. 2) or in three-dimensional (3D) floating collagen gels to generate mammary organoids (Fig. 1b). Following isolation, NMCs and LMCs from four donors each were examined for different mammary subpopulations using flow cytometry and a well-established panel of markers[17]. While the profiles of NMCs fit with previously identified populations[11], consisting of a CD45−/EpCAM−/CD49f+ basal MY, a CD45−/EpCAM+/CD49f+ LP and a CD45−/EpCAM+/CD49f− mature luminal subpopulation (Fig. 1c and Supplementary Fig. 1b–d), the same subpopulations were not clearly distinguishable in LMCs and were highly variable between participants (Fig. 1c, Supplementary Fig. 1b–d and Supplementary Table 1). Although many DRAQ5+ nucleated single LMCs stained positive for CD45 (4.5–43.7.9%, Fig. 1c and Supplementary Fig. 1b, c) the CD45− compartment did not display a clearly distinguishable MY subpopulation (0.2–1.0%, Fig. 1c and Supplementary Fig. 1b, c) nor revealed a clear distinction between EpCAM+ and EpCAM− cells (Fig. 1c and Supplementary Fig. 1b, c). Rather, a linear relationship between the expression of EpCAM and CD49f existed across the samples, indicating a potential loss of cell surface marker expression by these milk-derived cells. It is clear from this analysis that using only a few markers established for NMCs using flow cytometry is insufficient to characterize subpopulations existing in LMCs, hence we used single cell-transcriptomic analysis to determine the phenotypic differences between cells derived from these different maturation states.

To better define mammary cell subpopulations in human milk compared to resting breast tissue we profiled over 110,000 cells from nine LMC and seven NMC donors using single-cell RNA-sequencing (scRNA-seq; Fig. 1d and Supplementary Figs. 3 and 4, note some samples were sequenced twice). After batch correction, filtering and normalization (see "Methods" and Supplementary Fig. 3 for details), we found that individual sample or batch variation did not affect the global structure of the data (Fig. 1e and Supplementary Figs. 4–6), suggesting adequate correction was performed. Indeed, differences in sample preparation of LMCs (either using fresh milk or viably frozen LMCs) did not affect the resulting uniform manifold approximation and projection (UMAP) visualization of the data (Supplementary Fig. 5b, c). From these results, we concluded that separation between LMCs and NMCs was due to their origin (milk or non-lactating tissue, see principal component (PC) analysis in Supplementary Fig. 4b), rather than inter-donor variation and thus allowed us to probe the transcriptomic differences between the lactating and non-lactating human mammary gland.

### Two distinct secretory clusters characterize the luminal compartment in the lactating mammary gland. Mammary cell subpopulations were identified by conducting graph clustering which revealed 5 major epithelial cell clusters across all sequenced mammary cells. Among these, 3 clusters contained cells derived exclusively from NMCs which we found to represent a single MY cluster and two luminal clusters (LCs), in agreement with previous human mammary scRNA-seq studies[12–15]. Co-expression of established MY markers encoding for transcription factor p63 (TP63), keratin 17 (KRT17), metallopeptidase CD10 (MME), together with contractility genes encoding for alpha-smooth

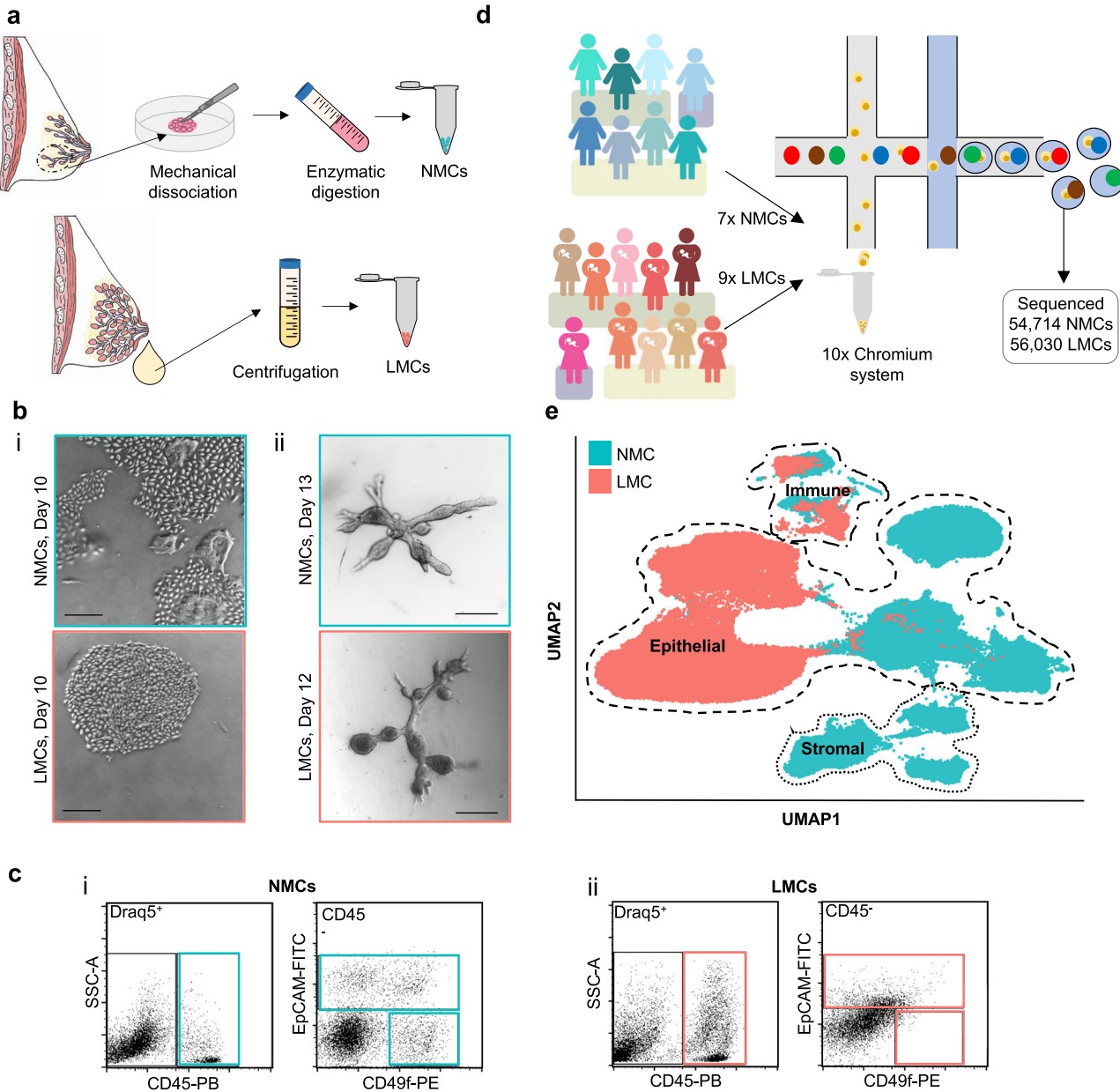

**Fig. 1 Exploring the diversity between non-lactating mammary cells (NMCs) and lactation derived mammary cells (LMCs). a** Cells from non-lactating tissue (above) and human milk (below) were isolated using either mechanical dissociation and enzymatic digestion or centrifugation, for downstream analysis. **b** Mammary cells from both non-lactating breast tissue (top) or lactating milk cells (bottom) were cultured either in **i** 2D (*n* = 10, see Supplementary Fig. 2) or **ii** 3D, scale bar represents 250 μm. **c** Representative flow cytometric profiles of immune/stromal (Draq5⁺/CD45⁺), luminal (Draq5⁺/EpCAM⁺/CD49f⁺/⁻) and myoepithelial cells (Draq5⁺/EpCAM⁻/CD49f⁺) from **i** NMCs and **ii** LMCs. **d** Schematic diagram for the scRNA-seq experimental set-up for cell samples from seven non-lactating participants and nine lactating females. **e** Uniform manifold approximation and projection (UMAP) dimensional reduction of the mammary cells reveals distinct clusters arising from NMCs and LMCs.

muscle actin (*ACTA2*), transgelin (*TAGLN*), myosin light chain kinase (*MYLK*) and tropomyosin (*TPM2*) demarked a single cluster as containing MY cells. The two remaining NMC clusters expressed key luminal markers encoding for keratin 18 (*KRT18*) and EPCAM (*EPCAM*). Upon closer examination, one cluster resembled the HR cluster previously described[12,13] which expressed genes encoding for hormone receptors for oestrogen, progesterone and prolactin (*ESR1*, *PGR*, *PRLR*; Fig. 2a, b and Supplementary Fig. 6d). The last LC resembled the previously annotated "hormone insensitive"[13] or "secretory L1"[12] clusters that co-expressed transcription factor *ELF5* as well as *ALDH1A3*, *KIT*, *SLPI* and *KRT23*. These markers are also characteristic of "LP" cells in the mouse[5,8], and hence for the purposes of this

study, we denote cells in this cluster as LP cells (Fig. 2a, b and Supplementary Fig. 6d). We found no significant differentially expressed genes (DEGs) in LP cells taken from parous compared to nulliparous individuals (Supplementary Fig. 7). NMCs contain all major epithelial cell subpopulations previously described in human and mouse mammary scRNA-seq studies.

For the LMC samples, our clustering analysis identified 2 major epithelial clusters that contained a heterogeneous contribution from all 9 LMCs donors and a very small proportion of NMCs (0.6–3.0% of total NMCs) from all (including nulliparous) donors (Fig. 1a, Supplementary Fig. 6c). Due to the fact that both of these clusters co-expressed luminal markers (*KRT18* and *EPCAM*), major human milk protein genes (*LALBA*, *CSN2* and *CSN3*[18])

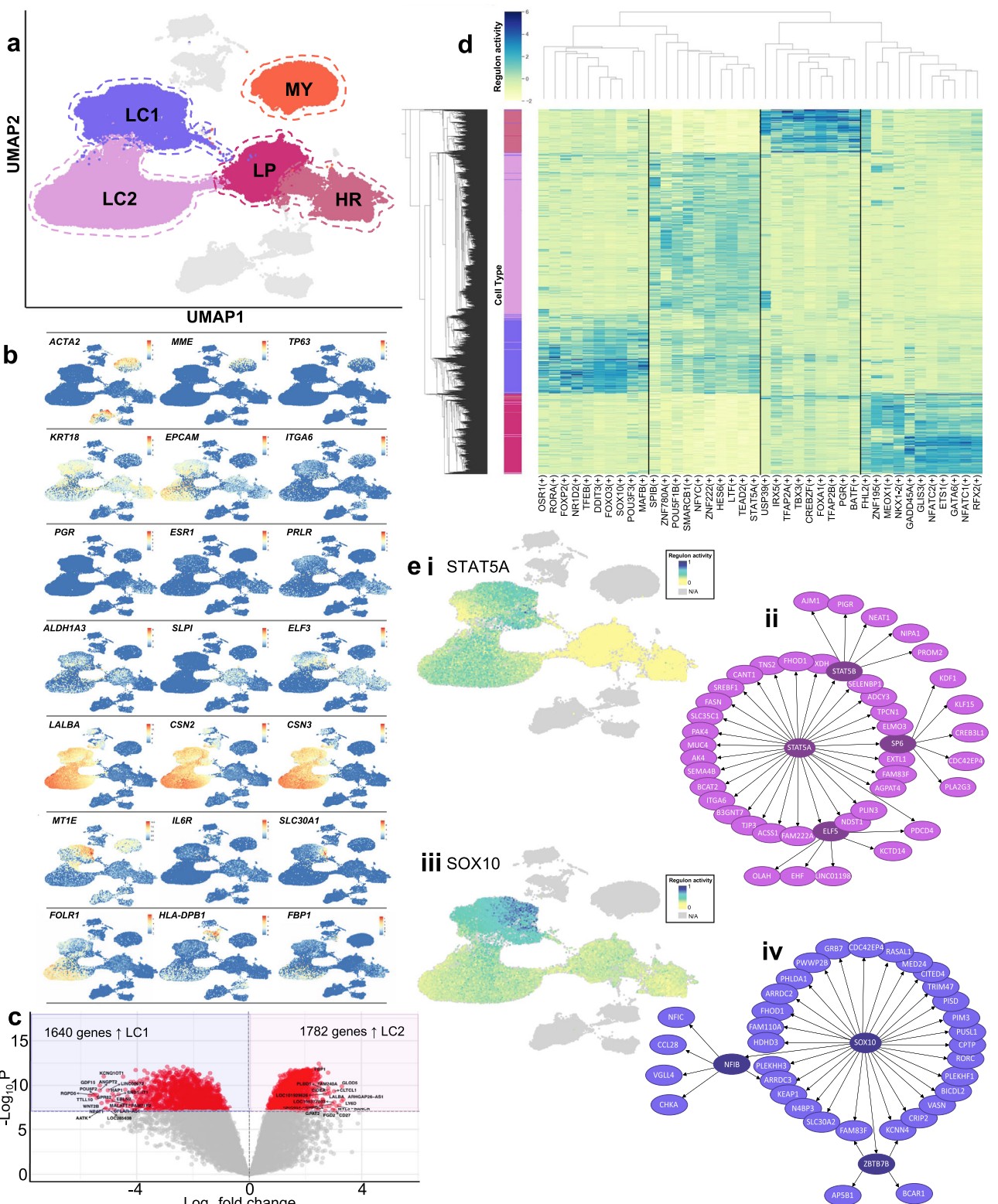

**Fig. 2 Clustering analysis of non-lactating (NMC) and lactation-associated (LMCs) mammary epithelial cells reveals different subpopulations arising from different developmental stages. a** Five major epithelial clusters were identified in our data set consisting of NMC myoepithelial (MY), luminal hormone-responsive (HR) and luminal progenitor (LP) clusters and LMC major luminal clusters 1 and 2 (LC1 and LC2). **b** Uniform manifold approximation and projections (UMAPs) coloured by marker genes characterizing the various clusters. **c** Volcano plot displaying the findings of the differential gene expression analysis revealed 1640 genes more highly expressed in LC1 compared to 1782 genes highly expressed in LC2. Significant genes are in red with the top 10 being annotated. **d** Top 10 regulons significantly upregulated in each luminal cell cluster. **e** Significant regulons found in LCs. **i** UMAP of STAT5A regulon activation in luminal cells and **ii** genes associated with STAT5A regulon. **iii** UMAP of SOX10 regulon activation in luminal cells and **iv** genes associated with SOX10 regulon.

and other secretory genes integral to milk fat secretion such as xanthine hydrogenase (*XDH*), CD36 (*CD36*) and mucin-1 (*MUC1*)[19] (Supplementary Fig. 6d), we designated these clusters as secretory LC clusters 1 and 2 (LC1 and LC2, Fig. 2a). We found the separation of cells into LC1 and LC2 to be highly reproducible across individuals (Supplementary Fig. 8). Within our analysis, we did not identify cells with gene expression profiles characteristic of other epithelial cell types (see Figs. 2a, b and 4a and Supplementary Figs. 6d and 9), in contrast to what has been described previously[20]. This is likely because our unbiased approach, which uses thousands of markers and bioinformatic tools to cluster cells based on similarity, overcomes the limitation of defining cell types based on only a handful of markers.

To better understand secretory luminal LMC heterogeneity, we compared LC1 to LC2 (Supplementary Fig. 10a) and found a total of 3422 genes significantly (false discovery rate (FDR) < $1 \times 10^{-8}$) differentially expressed, where 1640 genes were found to be higher in LC1 and 1782 genes were found to be higher in LC2 (Fig. 2c and Supplementary Dataset 1). Many of these genes were also identified when comparing LC1 and LC2 within the same individual (Supplementary Fig. 11). *FOLR1*, which encodes the alpha chain of the folate receptor (FR), was significantly upregulated in LC2 cells (Fig. 2b and Supplementary Dataset 1). FR could be used to separate viably frozen live LCs, which represented on average 46% (range: 17–81%) cells in milk, into two populations using flow cytometry analysis (Supplementary Fig. 10b). Significant genes upregulated in LC1 or LC2 were ordered according to their fold change differences and gene set enrichment analysis for gene ontology (GO) biological process terms was performed on the top 5% (82 and 89 genes, respectively) (Supplementary Fig. 10c and Supplementary Dataset 1–3). Overall, LC1 cells highly expressed genes associated with transcription, IM cell function and some biological process annotations indicating that the cells may be stressed (Supplementary Fig. 10ci, see Supplementary Dataset 2 for a full list). In contrast, LC2 highly expressed genes associated with lipid production and milk component biosynthesis, suggesting that these cells may represent a more mature secretory LC subtype (see Supplementary Fig. 10cii and Supplementary Dataset 3 for a full list). The fact that the two milk-derived LCs have such disparate gene expression profiles suggests not only potentially unique functions in the lactating mammary gland but also that LC1 and LC2 may be differentially regulated.

To understand differences in LC coordinated gene expression regulation, we performed single-cell regulatory network inference and clustering (SCENIC)[21] on a subset of each of the four LC clusters derived from NMCs and LMCs. We found a number of regulons that were highly expressed in each LC subtype (Fig. 2d and see Supplementary Dataset 4 for a full list), including previously noted[22] FOXA1 in HR cells and GATA6 in LP cells (Supplementary Fig. 12). Many of the top 10 regulons noted for LC2 were also highly expressed in LC1 cells (Fig. 2d and Supplementary Fig. 12), such as STAT5A (Fig. 2d, ei, eii), where the transcription factor *STAT5A* has been previously well established as essential for secretory LC differentiation and milk production[23–25]. In contrast, regulons such as SOX10 (Fig. 2eii) enriched for LC1 cells were found to be specific for this cluster. *SOX10* has been reported to be highly expressed in progenitor cells and identified as an important regulator of mammary gland development[26,27]. Together, this suggests that while LCs from milk share common highly expressed genes, they are most likely regulated by different transcription factors with different downstream effects. Overall, our data revealed that milk contains two distinct secretory cell populations that both highly express lactation-associated genes as well as gene expression profiles characteristic for each cell type that appear to be differentially regulated.

**Investigating non-lactating and human milk cell stromal (ST) and immune (IM) cell heterogeneity.** Our analysis identified five major clusters of ST and IM cells in the data set. In agreement with previous mammary scRNA-seq studies[13,15], we identified *GJPA4*+ (encoding gap junction protein alpha 4) vascular accessory (VA) cells, *PECAM1*+ (encoding CD31) endothelial (EN) cells and *DCN*+/*LUM*+/*COL1A1*+ fibroblasts (FB) within all NMC samples (Fig. 3a, b). Unsurprisingly, no VA, EN or FB lineage cells were isolated from any milk samples; however, all LMC and NMC samples contained cells belonging to the *PTPRC*+ (encoding CD45) IM cluster. To better determine the different subtypes of IM cells isolated from LMC and NMC samples, we performed sub-clustering analysis on the cells and annotated them according to the expression of canonical IM subpopulation markers[28]. Thus, we identified 12 subclusters consisting of either myeloid or lymphocytic lineage hematopoietic cells from NMC or LMC samples (Fig. 3b, c and Supplementary Fig. 13). One *CD68*+/*FCER1G*+ myeloid, three *ITGAX*+/*CD33*+ monocyte/ neutrophil and two *CD163*+/*MSR1*+/*C1QB*+ macrophage subclusters were identified consisting of LMC and NMC samples (Fig. 3b and Supplementary Fig. 13). Four clusters consisted of (*CD4*+, *IL7R*+) T cells or B cells (*CD79A*+/*MS4A1*+), with a small subset of *JCHAIN*+ plasma B cells consisting of both NMC and LMC samples (Fig. 3c and Supplementary Fig. 13). We noted that LMC-derived IM cells from all clusters also contained milk protein genes in their transcription profile, such as *LALBA*, *CSN2* and *CSN3* (Figs. 2b and 3a), likely due to ambient mRNA found in the milk and captured during the processing of cells for scRNA-seq.

To examine the potential signalling that occurs between IM cells and LCs in milk, we performed CellChat analysis[29] to infer putative heterotypic interactions (Fig. 3di). Interestingly, we found many potentially active signalling pathways between LC1/LC2 and the IM cell subtypes identified (Fig. 3dii and Supplementary Dataset 5). LC1, in particular, appeared to receive many signals from the IM cell compartment through the epidermal growth factor, midkine (member of the heparin growth factor family[30]) and osteopontin (SPP1) signalling pathways (Fig. 3diii and see Supplementary Fig. 14a for directionality) through multiple ligand–receptor signalling pairs (Supplementary Fig. 14). We found that both LC1 and LC2 expressed ligands and receptors for pathways typically associated with IM cell signalling such as major histocompatibility complex class I and II (MHC-I and MHC-II), colony-stimulating factor and granulin signalling (Fig. 3diii and Supplementary Fig. 14). These putative interactions suggest important feedback mechanisms, such as antigen presentation, from LCs to the surrounding IM cells during lactation. Together, this analysis reveals that, unlike the epithelial cell clusters, the IM LMCs mirror those in NMC (despite the differences in cell-isolation protocols), suggesting that the transcriptional differences we observe in the epithelium compartment reflect true biology.

**Exploring differences between milk-derived luminal clusters (LCs) and non-lactating luminal progenitors (LPs).** One major question arising from our data relates to the cell of origin of the milk-derived LCs. Which luminal population in the non-lactating, resting gland do they resemble most and likely arise from? To address this, we examined non-lactating cell signatures[31] derived from sorted mammary cell subpopulations[11] across all clusters, including LMCs. The purpose of this analysis was to take an unbiased approach and examine previously derived and curated gene signatures[31] from HR LCs ($n = 168$), LPs ($n = 169$ genes), MY ($n = 128$) or ST cells ($n = 384$) (Supplementary Dataset 6) in

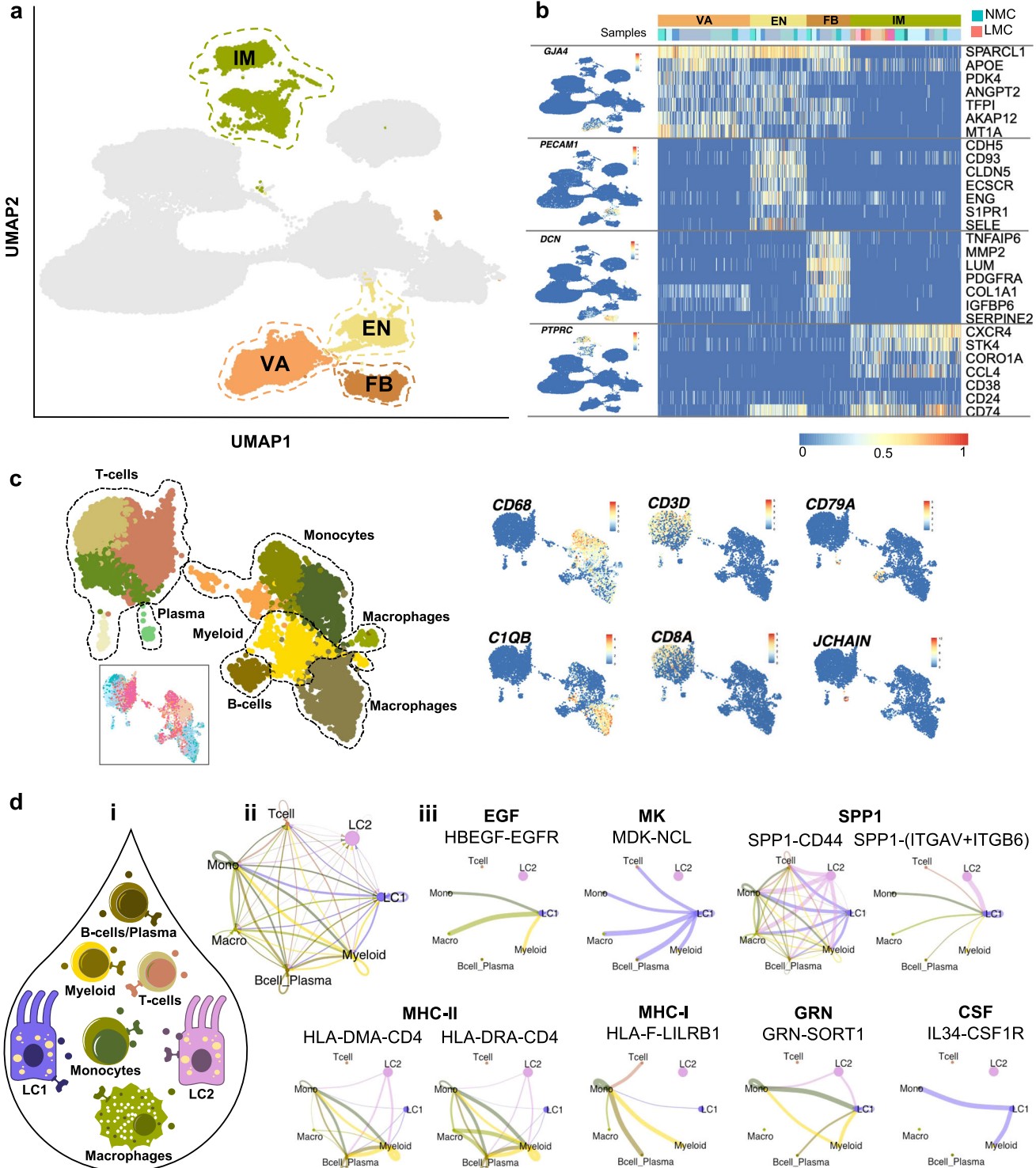

**Fig. 3 Investigation of the stromal compartment of non-lactating and lactating mammary epithelial cells. a** Stromal cells were classified into vascular accessory (VA), endothelial (EN), fibroblasts (FB) and immune (IM) cells. **b** Canonical stromal markers were used to classify the different stromal subtypes where LMC only contained IM cells. **c** Sub-setting and re-clustering of IM cells revealed that both myeloid and lymphocytic lineages were sequenced from both NMC and LMCs. **d** CellChat analysis between luminal and immune cell subtypes identified in milk (**i**). **ii** Observed interactions between the cell types. **iii** All immune cell subtypes signal to LC1 via the EGF, MK and SPP1 signalling pathway (selected receptor–ligand pairs shown), whereas LC1 and LC2 luminal clusters from milk are signalling to milk immune cells via MHC-II, MHC-I, CSF and GRN signalling pathways (selected receptor–ligand pairs shown).

our LC1 and LC2 clusters. Using these signatures, we calculated the combined expression level of these genes in each subpopulation resulting in a score that was visualized for each cluster (Fig. 4a). Reassuringly, each subpopulation signature was found to display the highest expression within our data set in the cluster(s) that we had independently assigned to the same subpopulation identity. The HR mature luminal signature was found to be highest in HR cells (Fig. 4ai), the LP score was highest in LP cells (Fig. 4aii), the MY score was highest in MY cells (Fig. 4aiii) and the ST score was highest across all ST clusters (Fig. 4aiv). These findings highlight the robustness of these gene signatures that include hundreds of markers and can be translated across bulk or scRNA-seq studies. It also revealed that both LC1 and more so LC2 displayed enrichment of the LP score, thus displaying a similar expression pattern to NMCs within the LP cluster (Fig. 4aii). These findings together with diffusion map (Supplementary Fig. 15) and UMAP (Fig. 2a) analysis, suggest that it is likely that secretory LCs arise from LPs in humans (as has been found in mice[5,8,9]). However, delineating exact differentiation pathways would require sampling mammary cells as they were differentiating (i.e. taken during pregnancy).

Next, we compared LP cells from NMC samples with both LMC secretory LCs LC1 and LC2 (collectively referred to as LC) using differential gene expression analysis. Thus, 1146 genes were found to be significantly upregulated in LC clusters and 922 genes were significantly upregulated in LP cells (FDR $< 1 \times 10^{-8}$, Fig. 4b and Supplementary Dataset 7). After ranking the significant DEGs by their fold change, we took a closer look at the top 10% (114 genes for LP and 92 genes for LC) and the GO biological process terms they were associated with. Genes found to be expressed at higher levels in LP NMCs were related to GO terms that could be broadly associated with changes to epithelial cell state, hormone response, cell trafficking, inflammation or cell adhesion (see Fig. 4b–d and Supplementary Dataset 8 for a full list of related GO terms). Interestingly, among the GO terms upregulated in LPs (and downregulation in LCs) was "cell adhesion" which suggests that LCs in milk may have downregulated many of their cell–cell adhesion molecules, either through an active process during lactation or due to being in suspension. Overall, GO terms associated with upregulated genes in LC LMCs were collectively related to fatty acid metabolism/storage, zinc transport, secretion and immunomodulatory response (see Fig. 4b–d and Supplementary Dataset 9 for detailed GO terms). Genes associated with similar terms have been previously found to be upregulated during lactation in mouse and human milk studies when comparing between different stages of development[5,32,33] but have not been previously examined in lactating human mammary epithelial cells.

Identification of genes such as these, likely to be involved in the normal function of the mammary gland, may in future be used to contrast those found during aberrant differentiation of the mammary gland during breast cancer. To this end, we compiled cell signatures for the major cell types in our samples including MY, LP, HR, LC1, LC2 and a collective ST signature (see Supplementary Dataset 10). Similar to the previous studies[11], we examined the expression of our cell signatures in tumour samples from The Cancer Genomics Atlas (TCGA) that were stratified based on the PAM50 molecular subtype classification[34]. As has been observed previously, the LP score was found to be highest in the "basal"-like tumour samples (Supplementary Fig. 16). This was also the case for the LC2 signature score. In contrast, the LC1 signature was not noticeably upregulated in any tumour subtype. Taken together, our results demonstrate that studying mammary cells from human milk provides insights into the function of the mammary gland and potentially leads to insights into breast cancer development.

## Discussion

Historically, it was thought that most of the epithelial cells found in human milk were simply dead or dying cells. We demonstrate here that these cells are alive. Specifically, we provide evidence to support the viability of milk-derived epithelial cells and indeed, that they can be maintained in vitro, similar to cells isolated from human breast tissue (Supplementary Figs. 2, 3 and 10). To better understand the biology of these cells, we performed scRNA-seq to compare for the first time differences in the composition of mammary cells isolated from human milk and non-lactating tissue. We identified two transcriptionally distinct secretory luminal epithelial cell clusters in milk samples from nine different donors. Our direct comparison to NMCs demonstrated that epithelial cells found in the milk transcriptionally resemble LP cells. In addition, this comparison also identified shared IM cell clusters from myeloid and lymphocytic lineages in both NMCs and LMCs. A recent study also identified epithelial and IM cells in milk samples from two donors[16]; however, a direct comparison to non-lactating tissue was not performed.

Curiously, we identified two secretory LC populations in our human samples (LC1 and LC2), unlike the single population that has been previously identified in mammary studies conducted in mice[5,8]. While both clearly display known and novel secretory genes, we find that additionally LC2 cells express high levels of immunomodulatory and antigen-presenting genes not previously associated with mammary LCs. The secreted proteins, through delivery to the infant, may play a role in the protection of the vulnerable newborn or indeed provide a mechanism for training the adaptive IM system of the infant[35]. This finding raises the question of why cells enter human milk. Some studies suggest that LMCs may enter into the milk for delivery to the infant and subsequently infiltrate into different organs for the benefit of the offspring[35–37], while others postulate that they enter due to a natural shedding process[38]. Live cells are likely to be secreted into milk by detaching (resulting in a downregulation of adhesion markers), entering either the lumen of the alveoli or ducts and being transported through the nipple to the breastfeeding infant. Considering the cellular organization of the mammary gland[2], it is therefore conceivable that cell subpopulations that reside distal to these sites (such as FBs or MY cells) do not enter the milk, compared to the more proximally localized LCs. Therefore, cells found in milk are unlikely to represent the full complement of populations in the lactating breast. Direct analysis of cells from human lactating breast tissue would be required to delineate further lactation-induced changes; however, obtaining such samples pose an ethical and logistical challenge.

The cell clusters identified in our NMC samples were in concordance with those reported in the previous studies[12–14,28]. Even though our NMC samples were taken from donors with a range of parities and ages, our analysis did not reveal major shifts in mammary cell populations influenced by these factors. Current efforts of the human breast cell atlas on very large numbers of genetically diverse participants will have more power to address questions around how age and parity affect the mammary cell composition[39]. Nonetheless, comparing NMCs to LMCs provided an insight into the potential epithelial differentiation trajectories that occur over mammary maturation. Comparisons between the LC LMCs and LP NMCs identified DEGs that provide clues into the maturation pathways modified during lactation, as well as potential genes involved in milk component biosynthesis pathways. These pathways may be further explored in vitro using mammary organoids and may be used to determine therapeutic targets for women with breastfeeding difficulties.

Finally, while mammary cells isolated from the milk do not directly contribute to breast cancer formation, they provide an opportunity to easily obtain and study human breast cells. These

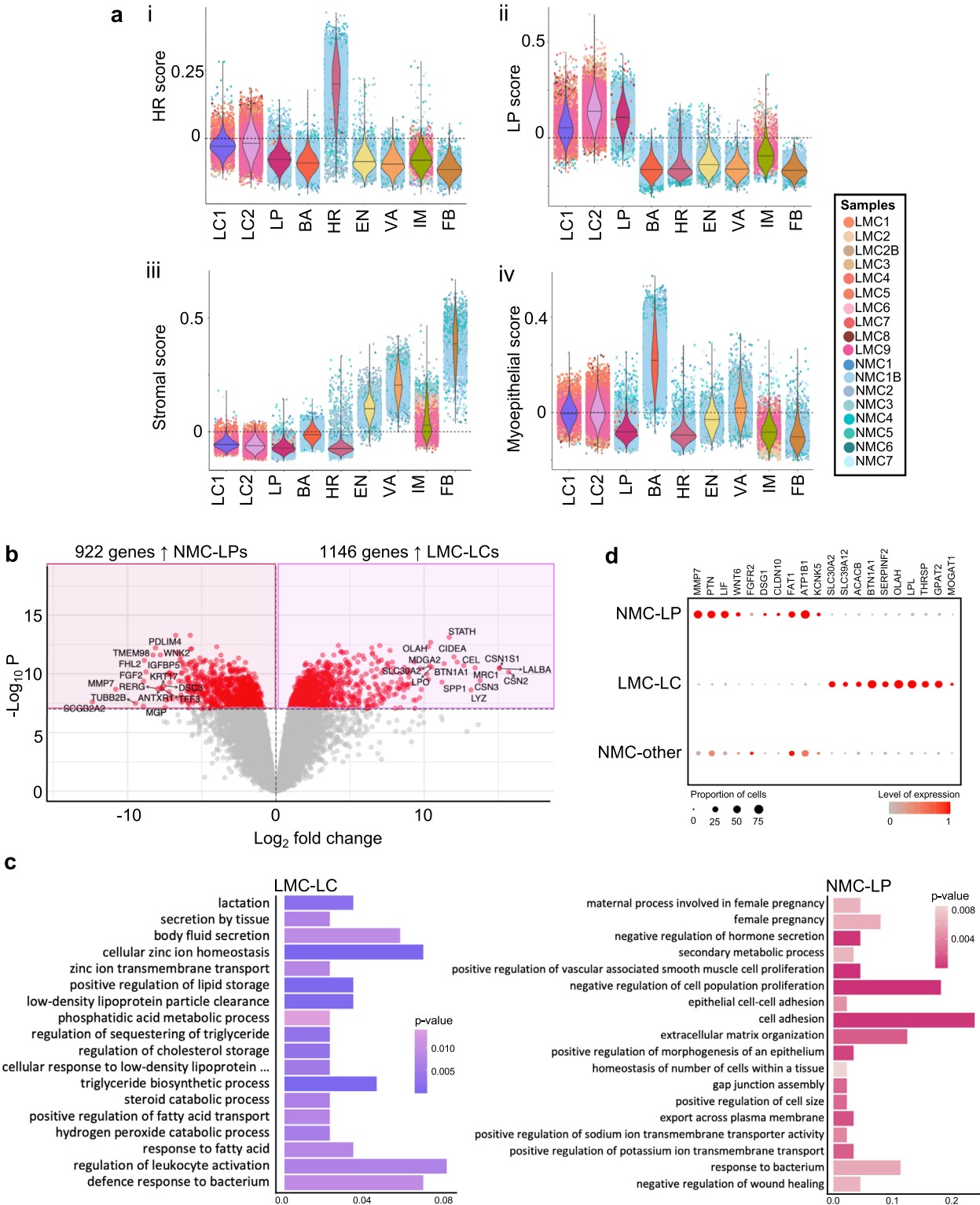

**Fig. 4 Comparing lactation-derived mammary cell (LMC) luminal clusters (LCs) with all other non-lactating mammary cell (NMC) types reveals similarity to non-lactating luminal progenitor (LP) cells. a** Violin plots of the mammary cell scores for **i** hormone-responsive (HR, mature luminal), **ii** luminal progenitor (LP), **iii** stromal or **iv** myoepithelial cells across the major cell clusters identified in this study. **b** Differential gene expression analysis derived using quasi-likelihood negative binomial generalized log-linear models revealed 1146 genes highly expressed in LC LMCs compared to 922 genes more highly expressed in LP NMCs, as displayed by a volcano plot (for a full list, see Supplementary Dataset 7). **c** Important biological process gene ontology pathways that were annotated by the top 10% of genes significantly differentially expressed (false discovery rate corrected $p$ value <1 × 10$^{-8}$) and upregulated in either LC (left) or LP (right); for a full list, see Supplementary Dataset 8 and 9. **d** Key LP (left) or LC (right) genes expressed in NMC-LP, LMC-LC or all other NMC clusters; colours represent overall normalized gene expression and size equals cell proportions.

cells could also allow us to better understand changes in the functional mammary gland over the course of lactation or as a result of multiple parities, if sampled longitudinally or over consecutive lactations. Furthermore, we have recently identified aberrant differentiation of LPs towards the milk secretory lineage as one of the earliest steps of tumorigenesis in a *BRCA1* mouse model[40], suggesting a hijacking of normal mammary cell programming during cancer, which, if better understood, maybe leveraged as an early indicator of future breast cancer development. Therefore, cells in the milk provide an opportunity to explore this phenomenon further and could aid in the development of a pre-clinical tool for early detection of disease. Together, our study demonstrates the power of comparing mammary cells isolated from different stages of human mammary gland maturation and illustrates the luminal lineage remodelling that occurs during lactation.

## Methods

Our research complies with all relevant ethical regulations as detailed in the below sections.

**Tissue collection and non-lactating cell isolation**. Non-lactating human breast tissue was donated by participants who provided informed consent who were undergoing elective aesthetic reduction mammoplasty at the Nymphenburg Clinic for Plastic and Aesthetic Surgery in accordance with the regulations of the ethics committee of the Ludwig-Maximilian University, Munich, Germany (proposal 397-12). Limited demographic information on the participants was provided that included the age and parity of the participant (Supplementary Fig. 4A). Single-cell suspensions of mammary cells were generated using an adaptation of a previously described protocol[41], however, in this case using a fast tissue digestion protocol (see Engelbrecht et al.[42]). Briefly, the freshly collected mammary tissue was collected and minced using scalpels into <2–3 mm³ pieces. Twenty millilitres of minced tissue in digestion buffer (Dulbecco's Modified Eagle Medium (DMEM)/F12 w/o phenol red, 2% w/v bovine serum albumin (BSA), 10 mM HEPES, 2 mM glutamine, 100 U/mL penicillin-streptomycin) supplemented with 1 µg/mL insulin (Sigma, I6634) was added together with 800 U/mL collagenase (Sigma, C9407) and 100 U/mL hyaluronidase (Sigma, H3506) and made up to the 25 mL mark. Falcon tubes were then sealed with parafilm to ensure they were airtight and mixed at 100 r.p.m. at 37 °C for 3–4 h. Following digestion, the resulting dissociated organoid fragments were washed twice in washing buffer (DMEM/F12 w/o phenol red, 10 mM HEPES, 2 mM glutamine, 100 U/mL penicillin–streptomycin) and pelleted by $300 \times g$ centrifugation for 5 min. Once the organoids were isolated, they were cryopreserved in 50% washing buffer, 40% foetal calf serum (FCS) and 10% dimethyl sulfoxide and stored in liquid nitrogen until required. When required, organoid fragments were gently defrosted in a 37 °C water bath for approximately 5 min before being treated with trypsin and dispase (Life Technology) to yield a highly viable single-cell suspension.

**Human milk cell isolation**. Human milk donors were recruited from Pippagina English Prenatal and Postnatal classes or through the Helmholtz Zentrum München in accordance with regulations of the ethics committee of the Ludwig-Maximilian University, Munich, Germany (proposal 17–715). Participants provided written informed consent and filled out a detailed questionnaire to provide demographic information. Briefly, human milk was freshly collected using either a provided double electric breast pump (Medela, Symphony) or participants personal pump, under aseptic conditions. Milk collections were obtained either within the lactation room at the Helmholtz Centre Munich, at the participants' homes or post-partum educational classes depending on the preferences of the participant. Fresh milk samples were immediately transported to the laboratory and processed as soon as possible (<2 h after collection). Human milk cells were isolated by diluting milk samples in an equal volume of sterile phosphate-buffered saline (PBS) (ThermoFisher Scientific, Waltham, U.S.) and centrifuged at $870 \times g$ for 20 min at 20 °C in a Rotanta 460R centrifuge (Hettich, Tuttlingen, Germany). The pellet was washed by removing the supernatant and resuspending in 5–10 mL of cold PBS before transferring the sample to a new 15 mL tube (Corning, Corning, U.S.) and centrifuging at $490 \times g$ for 5 min at 4 °C. Following a second washing step, 100–550 µL of mammary epithelial cell growth medium (MECGM) (PromoCell, Heidelberg, Germany) was added to the human milk cell aggregations according to the pellet size. Overall samples sequenced in this study ranged from 38 to 102.5 mL and yielded between $0.82 \times 10^6$ and $66.96 \times 10^6$ total membrane-enclosed structures (including cells) from milk (see Schultz-Pernice et al.[43] for more details). To examine the cells more closely, nuclear stain DRAQ5™ (62254, ThermoFisher Scientific, Waltham, U.S.) and Nile red (N3013-100MG, Merck, Darmstadt, Germany) were added to a final concentration of 0.4 µg/mL (1 mM) and 0.1 µg/mL, respectively, and the cells incubated for a further 5 min in the dark. Cells were then loaded onto a Neubauer Improved counting chamber and examined on an

immunofluorescence microscope. Subsequently, single cells were either frozen or used immediately.

**Cell culture**. Both NMC and LMCs were cultured in 2D and 3D using previously described methods[44]. Briefly, single cells were mixed with MECGM supplemented with 1% pen/strep (Invitrogen), 0.5% FCS (Pan Biotech), 3 µM Y-27632 (Biomol) and 10 µM forskolin (Biomol) and seeded onto polystyrene cell culture plates for 2D culture. After an establishment period of 5 days, the medium was changed to MECGM supplemented with 1% pen/strep and 10 µM forskolin. Where samples were cultured for immunofluorescence analysis, cells were seeded on 1% gelatine coated coverslips. Colonies of 2D-cultured cells were fixed using 4% paraformaldehyde and washed before staining the cells with a 1:100 primary CK8/18 antibody (DLN-010750, Dianova, Castelldefels, Spain) in blocking buffer (PBS 0.1% BSA (Merck, Darmstadt, Germany) with 10% normal donkey serum (GeneTex, Irvine, U.S.)) solution for 3 h at room temperature. Following washing, a 1:250 secondary antibody in (anti-mouse Alexa Fluor 488 Donkey Anti-Mouse IgG, A21202, ThermoFisher Scientific, Waltham, U.S.) PBS 0.1% BSA solution was added and samples were incubated for a further 45 min at room temperature. After antibody incubation, 500 µL of a 0.4 µg/mL DAPI solution (D9542-1MG, Merck, Darmstadt, Germany) were added and incubated for 2 min, followed by mounting of the slides. Images were acquired using a Zeiss "Axio Imager.M2" microscope in combination with the "Zeiss Zen 2.3 pro" software. For 3D culture, single cells were mixed with neutralizing solution and acidified rat tail collagen I (Corning) to generate collagen gels in siloxane coated 24-well plates. After allowing the gels to polymerize for an hour, supplemented media (as above) was added on top of the gels that were then gently encircled to generate floating collagen gels. Similar to 2D culture, after an establishment period of 5 days, the media on the floating collagen gels was changed to MECGM supplemented with pen/strep and forskolin only.

**Flow cytometry**. Flow cytometry was employed to determine the similarity of expression of mammary markers between LMC and NMCs. Cells were stained with CD45-PB (V450, dilution of 1:100, catalogue number: 560368, BD, Heidelberg, Germany), EpCAM-FITC (dilution of 1:10, catalogue number: GTX79849-100, GeneTex, Eching, Germany) and CD49f-PE (dilution of 1:20, BD, catalogue number: 555736, Heidelberg, Germany). After a 45-min incubation, stained cells were diluted in MECGM and filtered through 35 µm cell strainer caps of round-bottom tubes (Corning, Corning, U.S.). Ten minutes prior to analysis, cells were further stained with DRAQ5™ (as above) to a final concentration 1 µM. Flow cytometry was performed using a FACSAria™ III cell sorter (BD Biosciences, Franklin Lakes, U.S.) with a 100 µm nozzle in combination with the FACSDiva™ 6.0 Software. Alternatively, flow cytometry was performed on LMCs to examine FR expression in live, nucleated epithelial cells. LMCs were prepared by staining the cells with CD45-PB (dilution of 1:100, catalogue number: 304021, Biolegend, CA, U.S., catalogue number: 304021) and anti-human FRs α and β (FR-αβ)-PE (dilution of 1:40, catalogue number: 391805, Biolegend, CA, U.S., catalogue number: 391805) for 45 min in the dark on ice. Subsequently, the samples were stained for DAPI (prepared to 1 mg/mL, BD Pharmingen, Berkshire, U.K., catalogue number: 564907) and DRAQ5 (same as above) for 10 min prior to analysis. In this case, flow cytometry was performed on a BD LSRFortessa machine (BD Biosciences, Franklin Lakes, U.S.) in combination with the FACSDiva™ 9.0 Software. In both cases, small volumes of cells from each sample were mixed prior to staining and used as comparisons and single stain controls. Both machines were used with a 100 µm nozzle. Laser settings were adjusted using unstained and single stain controls. Obtained data were analysed using the FlowJo_V10 Software (FlowJo LLC, Ashland, U.S.).

**Library preparation, sequencing and data processing**. We examined cells that were prepared on three separate 10× genomics chips. Batch 1 contained four freshly dissociated NMCs (NMC1–4) and four samples of freshly collected and isolated LMCs (LMC1–4). Batch 2 contained three samples of NMCs (NMC5–7) and five viably frozen samples of LMCs (LMC5–8), including one sample that had been sequenced in the previous batch (LMC2B). Finally, batch 3 contained a new freshly collected and isolated LMC donor (LMC9) and a repeat sample of a freshly dissociated NMC donor (NMC1B), to examine batch effects (Supplementary Figs. 4a and 5a). Overall, we examined NMCs taken from nulliparous ($n = 3$) and parous ($n = 4$) females aged 19–65 years, as well as LMCs from uniparous or multiparous ($n = 5$ and $n = 4$, respectively) females aged 27–44 years, with lactation stages ranging from 2 to 12 months (Supplementary Fig. 4a). Library preparation for batch 1 and 2 was performed 10× Chromium single-cell kit using version 3 chemistry according to the instructions in the kit. The libraries were then pooled and sequenced on a NovaSeq6000 S2. Read processing was performed using the 10× Genomics workflow using the Cell Ranger Single-Cell Suite version 3.0.2. Samples were demultiplexed using barcode assignment and unique molecular identifier (UMI) quantification. The reads were aligned to the hg19 reference genome using the pre-built annotation package obtained from the 10× Genomics website (https://support.10xgenomics.com/single-cell-gene-expression/software/pipelines/latest/advanced/references). All lanes per sample were processed using the "cell ranger count" function. The output from different lanes was then aggregated using "cellranger aggr" with -normalise set to "none". Batch 3 was

prepared in a similar fashion, however, using version 2 chemistry, sequenced on an Illumina HiSeq 4000 and read processing was done using Cell Ranger Single-Cell Suite version 2.1.1.

**Quality control and data pre-processing**. Each batch had quality control and pre-processing performed separately using similar pipelines. Barcodes identified as containing low counts of UMIs likely resulting from ambient RNA were removed using the function "emptyDroplets" from the *DropletUtils* package[45]. For all batches, barcodes arising from single droplets were then filtered to ensure that cleaned barcodes contained at least 1000 UMIs and that the percentage of mitochondrial genes compared to overall annotated genes were not higher than 1× the median absolute deviation. Following filtering, expression values were normalized and log-transformed using the "computeSumFactors" from *scran* and "logNormCounts" from the *scater*[46] package.

Once each batch had undergone quality control and normalization the SingleCellExperiment objects were merged, intersecting genes kept and cells were normalized using the function "multiBatchNorm" from the *batchelor* package in R. Batch effects were removed by adding LMCs to NMCs by batch and performing mutual nearest neighbours (MNN) correction using the "fastMNN" function from the *batchelor* package. After performing quality control, combining, normalizing and performing MNN correction, we retained a total of high quality 54,714 NMCs and 56,030 LMCs (Fig. 1d and Supplementary Figs. 3 and 4) which had similar UMI, gene and % mitochondrial counts per cell, despite differences in cell source (Supplementary Fig. 3). Variance between the transcriptomic profile of single cells was examined by PC analysis using the "runPCA" function using *BiocSingular* version 1.6.0.

To visualize the global structure of the data, UMAP graphing (*umap* version 0.2.7.0) and Louvain clustering using *scran*[47] was performed on the batch corrected data (Fig. 1e). The resulting UMAP contained an even distribution of cells from each batch (Supplementary Fig. 5a) and sample (Supplementary Fig. 6a, b), suggesting that adequate correction was performed. In the case of the LMC and NMC samples that were repeatedly sequenced, we performed PC analysis and found no residual batch effects despite different sequencing days or in the case of LMCs whether the cells were prepared from freshly isolated or viably frozen cells (Supplementary Fig. 5c, d). Overall, 22 clusters were identified, which mapped to 9 major cell types including MY/basal, LP, HR, VA, FB, EN, IM and LC1 and LC2 (Supplementary Fig. 6b, c). We noted that the IM cell cluster appeared to have many subclusters, hence we performed subclustering analysis which revealed 12 clusters that were identified to represent 5 major IM cell subtypes based on gene expression profiles (Fig. 3c). Plots were generated using either *ggplot2* or *pheatmap* packages with custom colours generated by the *RcolourBrewer* package.

**Differential gene expression analysis**. DEGs were identified between subsetted groups by first generating pseudobulk samples using "aggregateAcrossCells" function in the *scater* package. *edgeR* version 3.34.3 was used to compute DEGs between groups by filtering and scaling sample library size using the "filterByExpr" and "calcNormFactors" functions. Next the common, trended and tagwise negative binomial dispersions of the genes were calculated using "estimateDisp". Quasi-likelihood negative binomial generalized log-linear models were fitted using "glmQLFit" and "glmQLFTest". FDR corrections were applied to the resulting $p$ values using the Benjamini–Hochberg method. To visualize the DEGs, volcano plots were generated using the *EnhancedVolcano* package the FDR corrected $p$ value cut off FDR $< 1 \times 10^{-8}$. Significant genes were ranked according to their FC and the top 5/10% of the positive (upregulated) or bottom 5/10% of the negative (downregulated) genes had gene set enrichment analysis performed on them using the "weight01" algorithm and "fisher" statistic using "runTest" in the "get-SigGroups" function from *topGO* package. "GenTable" was used to generate a table with the top 50 biological process GO terms. Plots of selected GO terms were generated using *ggplot2*, plotting the resulting $p$ value from the classic Fisher test and gene ratio, which is the number of significant genes for the term divided by the total number of significant genes used in the gene enrichment test.

**CellChat analysis**. CellChat analysis was performed using the R *CellChat* package version 1.1.0. First, LMC-derived cells were subsetted from the total cells and annotated according to their cell type: "LC1", "LC2", "Tcell", "Mono"(cytes), "Macro"(phages), "Bcell_plasma" or "Myeloid" cells. A CellChat object was made using the "createCellChat" function. After annotating the object with relevant labels and identifying overexpressed genes, the communication probability was inferred using "computeCommunProb" function. Cell–cell communications for each cell signalling pathway were generated with "computeCommunProbPathway" function. Graphs were generated using the "netVisual_chord_gene" function.

**Mammary cell signature score comparisons**. Mammary signatures[31] from previously published data[11] from LP, mature luminal, MY and ST cells were investigated in our data using the "AddModuleScore" function from the *Seurat* package[48] version 4.0.2. For each test, the overall expression of the genes/features from each signature was calculated after subtracting 20 randomly selected genes (from the same bin as the signature features) as a control feature per cell. The

resulting signature score is unitless but is indicative of signature enrichment per cell, which was then compared between clusters. Few genes in the published score were[47] not found in our data set and these have been reported in Supplementary Dataset 6.

**Cell signatures in TCGA tumours**. Cell signatures were derived from each cell subtype by using the "findMarkers" from *scran* package version 1.20.1. Genes that intersected between signature lists of multiple cell types were identified by the "intersect" function and were removed. Final gene lists for each cell type can be found in Supplementary Dataset 10. Samples were downloaded from the TCGA using the package *TCGAbiolinks* using the "GDCquery" function, where the parameters were set as follows project = "TCGA-BRCA", platform = "Illumina HiSeq", experimental.strategy = "RNA-Seq", sample.type = "c("Primary Tumor", "Solid Tissue Normal"). Only tumours that had been annotated for "paper_BR-CA_Subtype_PAM50" were kept for analysis ($n = 1083$). Cell signature scores were then examined in the TCGA data using the "AddModuleScore" function as described above.

**SCENIC analysis**. For the sake of computational speed, a random sample of 20,000 cells was selected from all luminal epithelial cells. The analysis was performed as previously described[21]. First, the gene regulatory network was constructed using "grn" function in pyscenic (version 0.11.1) with standard settings. The regulons were then identified using the "ctx" (--mask_dropouts) function. Finally, the area under the curve was computed using the "aucell" function with standard settings. The regulon specificity score was computed per regulon and cell type to receive a cell type-specific ranking of regulons. For the visualization of regulons (FIGREF to the graphs of e.g. SOX10), the top 30 downstream targets (ranked by "importance", see Supplementary Dataset 4 for all regulons) and top 5 secondary targets were plotted in a directed graph.

**Statistics and reproducibility**. No statistical methods were used to predetermine sample size. Unless stated otherwise, no data were excluded from the analyses. The experiments were not randomized and investigators were not blinded to allocation during experiments and outcome assessment.

**Reporting summary**. Further information on research design is available in the Nature Research Reporting Summary linked to this article.

## Data availability

The authors declare that all data supporting the findings of this study are available within the article and its Supplementary Information files or from the corresponding authors upon request. Each batch of the RNA sequencing data has been deposited in the Array Express database and can be retrieved by the following access IDs: "E-MTAB-9841" (Batch 1), "E-MTAB-10855" (Batch 2) and "E-MTAB-10885" (Batch 3), which will be released upon publication. A user-friendly website is available at http://bioinf.stemcells.cam.ac.uk:3838/khaled_wUFt1bHfmC/twigger/ to enable data exploration. All computational analyses were performed in R (Versions 3.5.3-4.1.0) using either standard functions or packages from Bioconductor (https://www.bioconductor.org/), downloaded through the program.

## Code availability

All codes used will be available online at https://github.com/aleciajane/LactatingMammaryCells.git.

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

## Acknowledgements

Many thanks are extended to Dr Anika Böttcher, Dr. Michael Sterr and Ms. Ines Kunze from the Helmholtz Diabetes Centre for assistance with preparing the samples for single-cell RNA-sequencing. Thank you to the Helmholtz sequencing core and particularly to Ms. Sandy Loesecke for conducting the sequencing. We thank Dr. Thomas Walzthoeni for bioinformatics support provided at the Bioinformatics Core Facility, Institute of Computational Biology, Helmholtz Zentrum München. Thank you to Hilary Ganz for her assistance with isolating cells for this study. Many thanks also to Katarzyna (Kasia) Kania and the Cancer Research United Kingdom Cambridge Institute (CRUK-CI) sequencing facility. This research was supported by the Cambridge NIHR BRC Cell Phenotyping Hub. This work was performed using resources provided by the Cambridge Service for Data Driven Discovery (CSD3) operated by the University of Cambridge Research Computing Service (www.csd3.cam.ac.uk), provided by Dell EMC and Intel using Tier-2 funding from the Engineering and Physical Sciences Research Council (capital grant EP/P020259/1), and DiRAC funding from the Science and Technology Facilities Council (www.dirac.ac.uk). Thanks to Dr. Mark Waterhouse for his support in setting up analysis on the aforementioned Cambridge high-performance computer (HPC). Thanks to Wellcome-MRC Cambridge Stem Cell Institute Bioinformatic core for their support with the data exploration website. A big thank you is extended to Ms. Lynn Darbyshire from Pippagina for her continued enthusiasm and support in collecting samples for this study and especially to all the women who provided their samples, without which this research would not be possible. A.-J.T. is funded by the ISRHML trainee bridge fund postdoctoral fellowship and the Helmholtz Postdoctoral Fellowship. K.B. is funded by a Cambridge Cancer Centre PhD studentship. This work was funded by a UKRI-MRC project grant ((MR/S036059/1), UKRI-BBSRC project grant (BB/S006745/1), Breast Cancer Now Project Grant (2017MayPR907) and a CRUK Programme Foundation Award (DCRPGF\100010) to W.T.K.

## Author contributions

A.-J.T. and C.H.S. conceptualized the study. A.-J.T., L.K.E., I.S.-P., S. Petricca and S. Pensa. collected samples and prepared samples for analysis. A.-J.T., I.S.-P. and J.P.S. performed flow cytometry and A.-J.T. and I.S.-P. performed culturing experiments with assistance from L.K.E. A.-J.T. performed bioinformatics analysis with assistance from K.B. A.-J.T., L.K.E., I.S.-P., K.B., J.S., S. Pensa, C.H.S. and W.T.K. interpreted the data. A.-J.T., C.H.S. and W.T.K. wrote the manuscript with critical input from all authors. C.H.S. and W.T.K supervised the study.

## Competing interests

The authors declare no competing interests.

## Additional information

**Peer review information** *Nature Communications* thanks Holly Holliday and the other anonymous reviewer(s) for their contribution to the peer review this work. Peer reviewer reports are available.

