## [Peer Review File · Nature Communications]

Reviewers' Comments:

Reviewer #1:

Remarks to the Author:

What are the noteworthy results?

Twigger et al. have performed single cell RNA-seq on cells isolated from human breast tissue and milk. The full set of epithelial and stromal populations were present in the tissue, while the milk cells contained secretory luminal cells and immune cells.

Two distinct clusters of luminal epithelial cells were observed from the milk: LC1 and LC2. LC2 had upregulation of transcripts related to antigen presentation and protein synthesis, while LC1 had an upregulated stress response.

The authors show that the luminal cells from milk most closely resemble luminal progenitors from the non-lactating breast, suggesting that they arise from luminal progenitors.

Genes involved in milk biosynthesis were identified by comparing luminal cells from milk to luminal progenitors from non-lactating breast tissue. This data will be a valuable resource to the mammary gland field due to the challenges in acquiring fully differentiated milk-secreting cells from humans.

Will the work be of significance to the field and related fields? How does it compare to the established literature? If the work is not original, please provide relevant references.

I think the work is of significance to the field of mammary gland biology. Single cell transcriptomics from murine and human breast cells have been enlightening for understanding differentiation dynamics during lineage commitment (Bach et al. 2017; Pal et al. 2017; Girardi et al. 2018; Nguyen et al. 2018). However, the breast does not complete its differentiation until lactation. This study builds upon existing literature by analysing terminally differentiated human breast cells.

The 2 distinct sub-clusters of luminal cells in milk are of potential interest, though at this stage the functional relevance is not clear. Are they different differentiate states along the same secretory trajectory, or do they represent distinct functional cell types?

It is well established from mouse models that luminal progenitor cells differentiate into secretory alveolar cells (Oakes et al. 2008; Rios et al. 2014; Bach et al. 2017). This study provides the first evidence that this is also the case in humans, albeit by linking the two cell types using two different sample types.

Does the work support the conclusions and claims, or is additional evidence needed?

Overall the work supports the conclusions, however I think there are some areas where this can be strengthened with further analysis and experiments.

1) The authors conclude that the LC cells come from LPs due to their similar transcriptome (Fig. 4a). Pseudotime analysis of single cells from both resting and lactating states should be performed. This may help strengthen the claim, and also help address if LC1 and LC2 are different differentiation states along the same lineage, or if they are distinct lineage branches.

2) The authors claim that hormone responsive luminal cells were missing from the milk, however in Fig 2b it appears that LC1 expresses ESR1 to a similar level to HR cells from breast tissue. Also, supplementary table 1 confirms significantly higher levels of ESR1 in LC1 compared to LC2. To test if LC1 cells are hormone responsive, the authors should perform TF regulon analysis (such as SCENIC (Aibar et al. 2017)) to see if ER targets are activated. This analysis should also reveal TFs important for driving lactation in LC2 such as STATs, and may even lead to the discovery of novel lactation driving TFs.

3) The two LC clusters (LC1 and LC2) (Fig. 2) could be explained by a population of stressed or dying cells, which may have shed from the epithelium early on before collection. As such they may not have any physiological significance. The fact that EPCAM expression is lost is suggestive that the cells are not happy (Fig. 1cii). Further validation of the LC1 and LC2 populations is needed. This could be achieved by performing scRNA-seq on the cultured cells/organoids derived from the milk to see if they are maintained in culture. Alternatively, flow cytometry of any cell surface markers identified in the original analysis would also be valid. This should be done for all donors.

4) The authors report that LC2 had upregulation of MHCII genes and suggest that these cells have a role in antigen presentation (Fig. 2c). I think the manuscript would benefit from investigating this further by performing cell-cell interaction analysis based on receptor-ligand interactions between the LC2 cells and the immune cells. There are several published tools available for this, for e.g. (Hou et al. 2020) and (Efremova et al. 2020).

Are there any flaws in the data analysis, interpretation and conclusions? - Do these prohibit publication or require revision?

I had some hesitations with some of the interpretations.

1) It is difficult to interpret the relative contribution of each donor in the current form in Fig. S2c. Could the authors please include a supplementary figure displaying 8 separate UMAP plots to ensure the reader that the cell clusters are representative of all donors.

2) To ensure the two LC clusters are not driven by inter-donor variation, the differential expression between LC1 and LC2 should be performed on a per-donor basis as a supplementary figure. If the individual analyses look representative of the integrated data, then it is fine to present the integrated analysis in the main figure.

3) While it is an interesting idea, I am slightly sceptical of the interpretation that immune cells have engulfed EVs containing milk transcripts as a signalling mechanism (Fig. 2b). It is more likely due to ambient RNA or cell doublets. To rule these possibilities in or out, FACS-sorting immune cells from breast milk and qPCR for milk transcripts should be performed.

4) Batch effects due to different collection procedures. I am worried that the different isolation methods have contributed to the most variation in gene expression. For example, in Fig. 3C, the CD8+ T cell clusters from the tissue and milk should cluster together but they do not. A control experiment whereby the milk cells are subjected to the same enzymatic digestion as the tissue cells would give confidence that the cells are indeed transcriptionally different. Alternatively, filtering out the artifactual milk transcripts from the T cells and re-clustering the data may address this.

Is the methodology sound? Does the work meet the expected standards in your field?

Other than the potential batch effects arising from the different collection procedures or individual differences, the experiments are carried out to the expected standards in the field.

Is there enough detail provided in the methods for the work to be reproduced?

Yes, the methods are sufficiently detailed to be repeated.

Minor points

- Define what DRAQ5 marks in the text
- CD31 mentioned in Figure 1 legend, but no mention in the figure or the text
- Typo line 153 PGR not PRG
- Line 179 should refer to supplementary 3b not 3c
- There is no mention in the text of Fig. 4d

- I found the acronyms confusing. The lactation derived mammary cells (LMC) and non-lactation derived mammary cells (NMC) are defined in the introduction. Then HMC and NLC are used in the figures (Fig. S2,3,4), and I couldn't find these defined in the text.
- An important missing cell type is the fully differentiated myoepithelial cells, this could be included in the discussion as a limitation.

References:

Aibar S, González-Blas CB, Moerman T, Huynh-Thu VA, Imrichova H, Hulselmans G, Rambow F, Marine JC, Geurts P, Aerts J et al. (2017). SCENIC: single-cell regulatory network inference and clustering. *Nature methods* 14: 1083-1086.

Bach K, Pensa S, Grzelak M, Hadfield J, Adams DJ, Marioni JC, Khaled WT. (2017). Differentiation dynamics of mammary epithelial cells revealed by single-cell RNA sequencing. *Nature communications* 8: 2128.

Efremova M, Vento-Tormo M, Teichmann SA, Vento-Tormo R. (2020). CellPhoneDB: inferring cell-cell communication from combined expression of multi-subunit ligand-receptor complexes. *Nature protocols* 15: 1484-1506.

Girardi RR, Chung C-Y, Heinz RE, Balcioglu O, Novotny M, Trejo CL, Dravis C, Hagos BM, Mehrabad EM, Rodewald LW et al. (2018). Single-Cell Transcriptomes Distinguish Stem Cell State Changes and Lineage Specification Programs in Early Mammary Gland Development. *Cell reports* 24: 1653-1666.e1657.

Hou R, Denisenko E, Ong HT, Ramilowski JA, Forrest ARR. (2020). Predicting cell-to-cell communication networks using NATMI. *Nature communications* 11: 5011.

Nguyen QH, Pervolarakis N, Blake K, Ma D, Davis RT, James N, Phung AT, Willey E, Kumar R, Jabart E et al. (2018). Profiling human breast epithelial cells using single cell RNA sequencing identifies cell diversity. *Nature communications* 9: 2028.

Oakes SR, Naylor MJ, Asselin-Labat ML, Blazek KD, Gardiner-Garden M, Hilton HN, Kazlauskas M, Pritchard MA, Chodosh LA, Pfeffer PL et al. (2008). The Ets transcription factor Elf5 specifies mammary alveolar cell fate. *Genes & development* 22: 581-586.

Pal B, Chen Y, Vaillant F, Jamieson P, Gordon L, Rios AC, Wilcox S, Fu N, Liu KH, Jackling FC et al. (2017). Construction of developmental lineage relationships in the mouse mammary gland by single-cell RNA profiling. *Nature communications* 8: 1627.

Rios AC, Fu NY, Lindeman GJ, Visvader JE. (2014). In situ identification of bipotent stem cells in the mammary gland. *Nature* 506: 322-327.

Reviewer #2:

Remarks to the Author:

The current manuscript addresses an aspect of human mammary gland biology that is currently poorly understood: lactation. The authors perform single cell RNA sequencing on human milk derived cells from 4 lactating donors and compare them with mammary epithelial cells isolated from non-lactating breast tissue.

Conceptually, the use of cells present in milk makes an indirect approach to study the lactating mammary gland, however, it is unclear whether these cells recapitulate the complexity of all the different cell subtypes present in a fully functional lactating gland, preventing the comparison between both sources and questioning the conclusions extracted in this work. Are these cells actually damaged cells that are being eliminated from the healthy tissue? The two sources of cells are not comparable.

The article is purely descriptive and not novel, as similar studies have been recently published: Single Cell RNA Sequencing of Human Milk-Derived Cells Reveals Sub-Populations of Mammary

Epithelial Cells with Molecular Signatures of Progenitor and Mature States: a Novel, Non-invasive Framework for Investigating Human Lactation Physiology.

Martin Carli JF, Trahan GD, Jones KL, Hirsch N, Rolloff KP, Dunn EZ, Friedman JE, Barbour LA, Hernandez TL, MacLean PS, Monks J, McManaman JL, Rudolph MC. J Mammary Gland Biol Neoplasia. 2020 Nov 20. doi: 10.1007/s10911-020-09466-z. Online ahead of print.PMID: 33216249.

Given that it would be impossible to isolate lactating tissue from humans , it would be interesting if the authors compared their findings to similar studies in other species, mouse, cow, ...but even then the conceptual advance to the field is unclear.

Reviewer #3:

Remarks to the Author:

Twigger et al performed single-cell RNA sequencing with mammary epithelial and immune cells isolated from human breast milk and non-lactating age-matched tissues. This study identified a common luminal compartment between both the sample types. They studied the difference between the luminal compartment to understand the differences in maturation states of these cells during resting state vs breastfeeding. The study is interesting. Obviously there are only 4 patients studied, and one parous, but an impressive number of cells profiled.

1. One of the major drawbacks of the study is they miss out on comparing the tissue signature from involuted breast and compare that to human breast milk derived cells and non-parous cells (only tissue derived from 1 parous female was used in the study). This could have made it better understand parity mediated impact on breast cancer risk. Moreover, provide better insights into how breastfeeding induced changes post lactation period as compared to breast tissue from non-parous individuals. perhaps this can be discussed.

2. It could be helpful to validate the finding that milk derived LP cells could be used to study early breast cancer in METABRIC/TCGA datasets

3. Number of cells analysed as mentioned in line 69 do not match the numbers defined methods section line 463. Please correct and also provide with QC information in supplementary, including number of genes/cell, fraction of mtRNA compared to all genes, number of genes/cluster/cell etc.

4. The authors need to define the abbreviations used to describe different samples types eg HMC LMC, MES which are otherwise hard to follow.

5. It seems surprising that human breast milk 10x did not capture B cells, whereas several studies have reported presence of B cells/plasma cells in milk and not just breast tissue. Can authors please comment on that.

6. Line 172-174: Can authors expand more on why no clusters of myoepithelial /stem cell origin where detected? Published studies have shown the presence of these subsets in human breast milk.

7. I can't seem to find all the table files as described within the manuscript.

8. Line 190: Bovine lactation? Sorry, I am missing the point here.

9. Line 234-237: I am confused that the clusters described here does not match what is observed in actuals figures. Those gene seem to be expressed in LC1 and LC2 clusters, myeloid and not of the immune clusters CD8. Can the authors show CD8 transcript level in that cluster?

10. Line 353 to 356 "Comparisons...low milk production" seems out of context/no in line with theme of the manuscript.

11. Can authors explain in the methods more on cell recovery from milk? How much milk is

needed to extract that many cells? Did they use a blocking buffer during staining? Esp since lactocytes are sticky and may cause background staining. Moreover, I did not see used of viability marker for FACS. Can the authors pls provide staining with isotype controls?

12. FACS ARIA nozzle is 100um and not in nm.

13. Line 515: there is typo in "were".

Response to reviewers' comments

The authors thank all the reviewers for their feedback. We believe that the reviewers provided insightful feedback which helped improve the quality and rigour of our study and conclusions. Below you will find the **original reviewers' comments in bold** and our *point-by-point response italicised below*.

REVIEWER COMMENTS

Reviewer #1 (Remarks to the Author):

What are the noteworthy results?

Twigger et al. have performed single cell RNA-seq on cells isolated from human breast tissue and milk. The full set of epithelial and stromal populations were present in the tissue, while the milk cells contained secretory luminal cells and immune cells.

Two distinct clusters of luminal epithelial cells were observed from the milk: LC1 and LC2. LC2 had upregulation of transcripts related to antigen presentation and protein synthesis, while LC1 had an upregulated stress response.

The authors show that the luminal cells from milk most closely resemble luminal progenitors from the non-lactating breast, suggesting that they arise from luminal progenitors

Genes involved in milk biosynthesis were identified by comparing luminal cells from milk to luminal progenitors from non-lactating breast tissue. This data will be a valuable resource to the mammary gland field due to the challenges in acquiring fully differentiated milk-secreting cells from humans.

Thank you for endorsing the importance of this resource to the mammary gland community.

Will the work be of significance to the field and related fields? How does it compare to the established literature? If the work is not original, please provide relevant references.

I think the work is of significance to the field of mammary gland biology. Single cell transcriptomics from murine and human breast cells have been enlightening for understanding differentiation dynamics during lineage commitment (Bach et al. 2017; Pal et al. 2017; Giraddi et al. 2018; Nguyen et al. 2018). However, the breast does not complete its differentiation until lactation. This study builds upon existing literature by analysing terminally differentiated human breast cells.

The 2 distinct sub-clusters of luminal cells in milk are of potential interest, though at this stage the functional relevance is not clear. Are they different differentiate states along the same secretory trajectory, or do they represent distinct functional cell types?

It is well established from mouse models that luminal progenitor cells differentiate into secretory alveolar cells (Oakes et al. 2008; Rios et al. 2014; Bach et al. 2017). This study provides the first evidence that this is also the case in humans, albeit by linking the two cell types using two different sample types.

Does the work support the conclusions and claims, or is additional evidence needed?

Overall the work supports the conclusions, however I think there are some areas where this can be strengthened with further analysis and experiments.

1) The authors conclude that the LC cells come from LPs due to their similar transcriptome (Fig. 4a). Pseudotime analysis of single cells from both resting and lactating states should be performed. This may help strengthen the claim, and also help address if LC1 and LC2 are different differentiation states along the same lineage, or if they are distinct lineage branches.

We thank the reviewer for their comment. For the revision we have now sequenced samples from 9 additional donors (6 milk cells and 3 breast tissue cells). As part of integrating the datasets we applied computational batch correction methods (see response to major point 4). In this larger and batch corrected dataset we find that epithelial cells in milk do appear to be closely related to LPs in the resting breast tissue (see UMAP in Figure 2a and diffusion map in Supplementary Figure 12). As can be seen via dimensional reduction analysis, LC2 cells are closely related to LPs. When we perform pseudotime analysis on LPs and LCs, we find that LC1 seems to be more closely related to LPs. The pseudotime analysis also highlights a gap between LP and LC1 which is expected as cells from the intermediate developmental stage, gestation, are not part of our dataset. This is discussed in the main text at lines 258-262.

2) The authors claim that hormone responsive luminal cells were missing from the milk, however in Fig 2b it appears that LC1 expresses ESR1 to a similar level to HR cells from breast tissue. Also, supplementary table 1 confirms significantly higher levels of ESR1 in LC1 compared to LC2. To test if LC1 cells are hormone responsive, the authors should perform TF regulon analysis (such as SCENIC (Aibar et al. 2017)) to see if ER targets are activated. This analysis should also reveal TFs important for driving lactation in LC2 such as STATs, and may even lead to the discovery of novel lactation driving TFs.

We thank the reviewer for this insightful suggestion. We have now performed the regulon analysis on all luminal cell clusters captured in our larger dataset using SCENIC, as suggested by the reviewer. We found a number of significantly enriched regulons in each luminal cell cluster (see Figure 2d-e and Supplementary Figure 9 and Supplementary Table 4). Of note, STAT5A was enriched in both LC1 and LC2 while the SOX10 regulon is enriched in LC1 only. We did not find an enrichment for ESR1 in any of the luminal subtypes (Reviewer Figure 1), however, we do find enrichment for the ER pioneer factor FOXA1 in hormone sensing cells (Reviewer Figure 1, Supplementary Figure 9). The text is now also updated to include this new data, see lines 180-196.

Reviewer Figure 1: ESR1 and FOXA1 regulon activity across the subset of luminal cells examined in the analysis. Low activity is marked in yellow, whereas high regulon activity within a cell is marked in dark purple.

3) The two LC clusters (LC1 and LC2) (Fig. 2) could be explained by a population of stressed or dying cells, which may have shed from the epithelium early on before collection. As such they may not have any physiological significance. The fact that EPCAM expression is lost is suggestive that the cells are not happy (Fig. 1cii). Further validation of the LC1 and LC2 populations is needed. This could be achieved by performing scRNA-seq on the cultured cells/organoids derived from the milk

to see if they are maintained in culture. Alternatively, flow cytometry of any cell surface markers identified in the original analysis would also be valid. This should be done for all donors.

We thank the reviewer for the comment. We could not perform the scRNAseq on cultured cells due to the limited material. Therefore, we decided to focus on identifying the cell surface markers that can be used to distinguish between LC1 and LC2. By mining the scRNAseq data we found some candidate cell surface proteins and then looked to see which of them have validated FACS antibodies that we can use. This led to one cell surface marker, the alpha chain folate receptor (FOLR1) (Figure 2b). To ensure that we are analysing only live epithelial cells, we examined nucleated (DRAQ5+), live (DAPI-), non-immune (CD45-) cells for FOLR1 expression which we found to be predominantly upregulated in LC2 but not LC1 cells (Supplementary Figure 7 and Supplementary Table 1). Due to the limited material available we ran the FACS analysis on four samples which were taken from the two largest batches. See lines 164-165 and 420-433 for in text discussion.

4) The authors report that LC2 had upregulation of MHCII genes and suggest that these cells having a role in antigen presentation (Fig. 2c). I think the manuscript would benefit from investigating this further by performing cell-cell interaction analysis based on receptor-ligand interactions between the LC2 cells and the immune cells. There are several published tool available for this, for e.g. (Hou et al. 2020) and (Efremova et al. 2020).

The authors thank the reviewer for this comment which has led to some very interesting data and provides more context to the potential interactions between mammary epithelial and immune cells during lactation. To identify putative heterotypic interactions from our scRNAseq data we used the CellChat algorithm (Jin et al. 2021). As outlined in lines 218-233, Figure 3d and Supplementary Figure 11, we found that many ligands and receptors were associated with the MHC-II pathway. In addition to the genes HLA-DRA and HLA-DPB1 noted in our original submission, we found many other genes enriched in immune related pathways such as MHC-I, CSF and GRN and hormone related pathways such as EGF and MZ (see Figure 3d, Supplementary Figure 11 and Supplementary Table 5).

Are there any flaws in the data analysis, interpretation and conclusions? - Do these prohibit publication or require revision?

I had some hesitations with some of the interpretations.

1) It is difficult to interpret the relative contribution of each donor in the current form in Fig. S2c. Could the authors please include a supplementary figure displaying 8 separate UMAP plots to ensure the reader that the cell clusters are representative of all donors.

This is a fair comment and we have provided individual UMAPs for all 18 samples analysed in this study in Supplementary Figure 4a, as well as the overall number of cells per sample (Supplementary Figure 2a) and relative contribution of each sample to each cell type (Supplementary Figure 4c). Also see lines 111-116 for in text references.

2) To ensure the two LC clusters are not driven by inter-donor variation, the differential expression between LC1 and LC2 should be performed on a per-donor basis as a supplementary figure. If the individual analyses look representative of the integrated data, then it is fine to present the integrated analysis in the main figure.

According to the reviewer's suggestion, we subset the luminal cells for each donor and subsequently performed principal component analysis on the cells. We found that for each participant, the cells separated into two major clusters, which upon colouring the cells by their original annotation of LC1 and LC2 (Supplementary Figure 5), we found that for each donor, luminal cells separate into LC1 and

LC2. If the reviewer is interested in the relative contribution from each donor to LC1 and LC2, a diagram demonstrating this can be found in Supplementary Figure 4c. Further, according to the suggestion by the reviewer, we examined genes that were differentially expressed between LC1 and LC2 per sample using MA plots (Supplementary Figure 8). As can be seen from these plots, many of the genes found to be differentially expressed between LC1 and LC2 across all samples (see Supplementary Table 1) were also found in the individual sample comparisons between LC1 and LC2. We therefore conclude that the integrated analysis between LC1 and LC2 is representative of differences found on an individual level too.

3) While it is an interesting idea, I am slightly sceptical of the interpretation that immune cells have engulfed EVs containing milk transcripts as a signalling mechanism (Fig. 2b). It is more likely due to ambient RNA or cell doublets. To rule these possibilities in or out, FACS-sorting immune cells from breast milk and qPCR for milk transcripts should be performed.

The authors thank the reviewer for an excellent point. When we prepared the extra samples for sequencing, we added in spike-in cells (human mammary epithelial cells, HuMECs, from Thermo Fisher with catalogue number: A10565 and lot number 2098293) to each sample that could be used as a control to help assess the levels of ambient RNA captured by our 10x runs. To deconvolute the spike in cells from the samples, we genotyped the samples and found that the spike-in cells contained many milk protein transcripts (see Reviewer Figure 2). Based on these findings, we agree that the milk transcripts found in the immune cells are highly likely derived from ambient RNA rather than from immune cells engulfing EVs. To reflect this shift in perspective, we have updated the manuscript accordingly (see lines 215-218).

Reviewer Figure 2: Spike in cells added into the lactating mammary cell (LMC) samples reveal that ambient milk protein RNA is captured in all LMC derived cells. a) UMAP of cells from our new batch of 8 samples containing 3 non-lactation associated mammary cells (NMCs) and 5 LMCs. **b)** UMAPs coloured by the results of genotyping our samples (by SNP analysis) reveals which cells are the spike ins and which are the cells from our donors. **c)** UMAPs from our new samples (and spike ins) coloured by milk protein gene transcripts.

4) Batch effects due to different collection procedures. I am worried that the different isolation methods have contributed to the most variation in gene expression. For example, in Fig. 3C, the CD8+ T cell clusters from the tissue and milk should cluster together but they do not. A control experiment whereby the milk cells are subjected to the same enzymatic digestion as the tissue cells would give confidence that the cells are indeed transcriptionally different. Alternatively, filtering out the artifactual milk transcripts from the T cells and re-clustering the data may address this.

We thank the reviewer for the comment. Given the limited precious material we have from each milk sample we reasoned that exposing milk cells, which are already in suspension, to collagenase and hyaluronidase is likely to be very toxic. However, we tackled this problem computationally as part of including the extra samples. We have now performed a batch correction on the larger dataset. In the updated figures (Figures 1e, 2a and 3a, c), it can be seen that the batch correction eliminated this artifact described by the reviewer. In the updated dataset, the dimensionality reduction separates immune cells based on whether they are derived from lymphocytic or myeloid lineage cells rather than if they are derived from breast tissue or milk. Therefore, we believe that we have adequately addressed the reviewers concerns of batch effects derived from cell type by performing mutual nearest neighbours' (MNN) correction which accounts for these differences. See description of the methods lines 473-475 and updated results section lines 111-113.

Is the methodology sound? Does the work meet the expected standards in your field?

Other than the potential batch effects arising from the different collection procedures or individual differences, the experiments are carried out to the expected standards in the field.

We are grateful for the reviewer highlighting the batch correction issue which has now been addressed in the new analysis.

Is there enough detail provided in the methods for the work to be reproduced?

Yes, the methods are sufficiently detailed to be repeated.

Many thanks.

Minor points

- **Define what DRAQ5 marks in the text**

✓ See addition of "nuclear stain" in line 391.

- **CD31 mentioned in Figure 1 legend, but no mention in the figure or the text**

✓ This has been corrected, see Figure 1 legend.

- **Typo line 153 PGR not PRG**

✓ See line 136.

- **Line 179 should refer to supplementary 3b not 3c**

The figures have been dramatically changed to reflect the suggestions of the reviewers; therefore this has been updated.

- **There is no mention in the text of Fig. 4d**

This has been rectified.

- **I found the acronyms confusing. The lactation derived mammary cells (LMC) and non-lactation derived mammary cells (NMC) are defined in the introduction. Then HMC and NLC are used in the figures (Fig. S2,3,4), and I couldn't find these defined in the text.**

✓ These terms have been updated to only include defined terms.

- **An important missing cell type is the fully differentiated myoepithelial cells, this could be included in the discussion as a limitation.**

This has previously been discussed in detail in a referenced review, see Twigger and Khaled 2021.

References:

- Aibar S, González-Blas CB, Moerman T, Huynh-Thu VA, Imrichova H, Hulselmans G, Rambow F, Marine JC, Geurts P, Aerts J et al. (2017). SCENIC: single-cell regulatory network inference and clustering. *Nature methods* 14: 1083-1086.
- Bach K, Pensa S, Grzelak M, Hadfield J, Adams DJ, Marioni JC, Khaled WT. (2017). Differentiation dynamics of mammary epithelial cells revealed by single-cell RNA sequencing. *Nature communications* 8: 2128.
- Efremova M, Vento-Tormo M, Teichmann SA, Vento-Tormo R. (2020). CellPhoneDB: inferring cell-cell communication from combined expression of multi-subunit ligand-receptor complexes. *Nature protocols* 15: 1484-1506.
- Girardi RR, Chung C-Y, Heinz RE, Balcioglu O, Novotny M, Trejo CL, Dravis C, Hagos BM, Mehrabad EM, Rodewald LW et al. (2018). Single-Cell Transcriptomes Distinguish Stem Cell State Changes and Lineage Specification Programs in Early Mammary Gland Development. *Cell reports* 24: 1653-1666.e1657.
- Hou R, Denisenko E, Ong HT, Ramilowski JA, Forrest ARR. (2020). Predicting cell-to-cell communication networks using NATMI. *Nature communications* 11: 5011.
- Nguyen QH, Pervolarakis N, Blake K, Ma D, Davis RT, James N, Phung AT, Willey E, Kumar R, Jabart E et al. (2018). Profiling human breast epithelial cells using single cell RNA sequencing identifies cell diversity. *Nature communications* 9: 2028.
- Oakes SR, Naylor MJ, Asselin-Labat ML, Blazek KD, Gardiner-Garden M, Hilton HN, Kazlauskas M, Pritchard MA, Chodosh LA, Pfeffer PL et al. (2008). The Ets transcription factor Elf5 specifies mammary alveolar cell fate. *Genes & development* 22: 581-586.
- Pal B, Chen Y, Vaillant F, Jamieson P, Gordon L, Rios AC, Wilcox S, Fu N, Liu KH, Jackling FC et al. (2017). Construction of developmental lineage relationships in the mouse mammary gland by single-cell RNA profiling. *Nature communications* 8: 1627.
- Rios AC, Fu NY, Lindeman GJ, Visvader JE. (2014). In situ identification of bipotent stem cells in the mammary gland. *Nature* 506: 322-327.

Reviewer #2 (Remarks to the Author):

The current manuscript addresses an aspect of human mammary gland biology that is currently poorly understood: lactation. The authors perform single cell RNA sequencing on human milk derived cells from 4 lactating donors and compare them with mammary epithelial cells isolated from non-lactating breast tissue.

Conceptually, the use of cells present in milk makes an indirect approach to study the lactating mammary gland, however, it is unclear whether these cells recapitulate the complexity of all the different cell subtypes present in a fully functional lactating gland, preventing the comparison between both sources and questioning the conclusions extracted in this work. Are these cells actually damaged cells that are being eliminated from the healthy tissue? The two sources of cells are not comparable.

The article is purely descriptive and not novel, as similar studies have been recently published:

Single Cell RNA Sequencing of Human Milk-Derived Cells Reveals Sub-Populations of Mammary Epithelial Cells with Molecular Signatures of Progenitor and Mature States: a Novel, Non-invasive Framework for Investigating Human Lactation Physiology. Martin Carli JF, Trahan GD, Jones KL, Hirsch N, Rolloff KP, Dunn EZ, Friedman JE, Barbour LA, Hernandez TL, MacLean PS, Monks J, McManaman JL, Rudolph MC. *J Mammary Gland Biol Neoplasia*. 2020 Nov 20. doi: 10.1007/s10911-020-09466-z. Online ahead of print. PMID: 33216249.

Given that it would be impossible to isolate lactating tissue from humans , it would be interesting if the authors compared their findings to similar studies in other species, mouse, cow, ...but even then the conceptual advance to the field is unclear.

The authors thank the reviewer for taking the time to review their manuscript. It is important to note that our study was submitted and available on BioRxiv before the publication of Martin Carli et. al which reports on the scRNA-sequencing of milk cells from 2 women. Nonetheless, our study addresses different questions to the referenced study. In this, updated manuscript, we have sequenced milk cells from 9 women and compared their transcriptional profile to breast tissue cells from 7 non-lactating women. We have previously published scRNAseq analysis of the mouse mammary gland across different developmental stages (Bach et al 2017) but we are not aware of similar datasets for cow or sheep. Whilst there might be value in cross-species comparison as the reviewer suggest this current study is focusing on the comprehensive analysis of the milk cells and comparing them to their counterparts from non-lactating human tissue. Therefore, we believe the cross-species comparison analysis is beyond the scope of this manuscript.

Reviewer #3 (Remarks to the Author):

Twigger et al performed single-cell RNA sequencing with mammary epithelial and immune cells isolated from human breast milk and non-lactating age-matched tissues. This study identified a common luminal compartment between both the sample types. They studied the difference between the luminal compartment to understand the differences in maturation states of these cells during resting state vs breastfeeding. The study is interesting. Obviously there are only 4 patients studied, and one parous, but an impressive number of cells profiled.

1. One of the major drawbacks of the study is they miss out on comparing the tissue signature from involuted breast and compare that to human breast milk derived cells and non-parous cells (only tissue derived from 1 parous female was used in the study). This could have made it better understand parity mediated impact on breast cancer risk. Moreover, provide better insights into how breastfeeding induced changes post lactation period as compared to breast tissue from non-parous individuals. perhaps this can be discussed.

The authors thank the reviewer for this valuable suggestion. After careful consideration of their point, we decided to further profile 3 more parous breast tissue and 6 more milk cell samples (see Figure 1a, Supplementary Figure 2a). We subsequently performed differential gene expression analysis between LPs from nulliparous and parous NMC samples (Reviewer Figure 3a) as well as between LCs from uniparous and multiparous LMCs (Reviewer Figure 3b). We found no significantly differentially expressed genes between the samples in either case. We also compared nulliparous LPs with all LCs, as well as all parous LPs with all LCs. As expected in both cases, we found many DE genes, however when we compared the fold changes of the DE genes, they were very similar (Reviewer Figure 3c). We believe that the number of samples we have analysed in this study is not sufficient to identify the changes that occur between from nulliparity to lactation to involution, however, will be able to make future comparisons between our data and larges datasets expected to come out of the human cell atlas (van Amerongen 2021).

Reviewer Figure 3: Examining the effect of parity on luminal cell gene expression. a) Volcano plot of the genes differentially expressed between nulliparous and parous luminal progenitors (LPs). **b)** Volcano plot of genes differentially expressed between uniparous and multiparous luminal cells (LCs) from milk. **c)** Contrasting fold changes of differentially expressed genes identified by comparing all LCs to either nulliparous or parous LPs.

2. It could be helpful to validate the finding that milk derived LP cells could be used to study early breast cancer in METABRIC/TCGA datasets.

We thank the reviewers for this suggestion. We decided to generate signatures for each major cell type, similar to what has been done previously (Lim et al. 2009) and examined expression of these signatures in tumour samples uploaded to the cancer genomics atlas (TCGA). These samples had been previously categorised by PAM50 molecular subtypes (see Supplementary Figure 13) from which we found that as with the LP cell signature, basal-like tumours have an upregulated LC2 signature. See further details of the methods in lines 538-548 and discussion in lines 284-296.

3. Number of cells analysed as mentioned in line 69 do not match the numbers defined methods section line 463. Please correct and also provide with QC information in supplementary, including number of genes/cell, fraction of mtRNA compared to all genes, number of genes/cluster/cell etc.

The authors thank the reviewer for spotting this error. This has been corrected and quality control graphs displaying the number of genes, unique molecular identifiers (UMIs) and percentage of mitochondrial counts per cell per sample have been added (see Supplementary Figure 2b-d). Additionally, the relative number of cells contributed to each cell type by each sample has been displayed in Supplementary Figure 4c.

4. The authors need to define the abbreviations used to describe different samples types eg HMC LMC, MES which are otherwise hard to follow.

✓ This has been rectified.

5. It seems surprising that human breast milk 10x did not capture B cells, whereas several studies have reported presence of B cells/plasma cells in milk and not just breast tissue. Can authors please comment on that.

Thank you for the insightful comment. Indeed, now after sequencing more LMCs we do find B-cells in milk (see UMAPs Figure 3c). We believe that we were unable to identify them in the original analysis because they appear to only make up a very small fraction of the total immune cell population.

6. Line 172-174: Can authors expand more on why no clusters of myoepithelial /stem cell origin were detected? Published studies have shown the presence of these subsets in human breast milk.

This is an excellent observation. We believe that a limitation of previous studies is that they often characterise cell subpopulations based on limited numbers of markers. We believe that scRNA-seq is unbiased and we have included cells from breast tissue which provides an important control for this analysis. In this way, we have found that markers that have been previously used to identify myoepithelial/stem cell populations in milk are in fact minimally expressed when compared to different subpopulations of human breast tissue (see Supplementary Figure 6). We have attempted to clarify this in lines 153-157.

7. I can't seem to find all the table files as described within the manuscript.

Reference to tables have now been carefully checked and where appropriate edited.

8. Line 190: Bovine lactation? Sorry, I am missing the point here.

This sentence has been removed.

9. Line 234-237: I am confused that the clusters described here does not match what is observed in actuals figures. Those gene seem to be expressed in LC1 and LC2 clusters, myeloid and not of the immune clusters CD8. Can the authors show CD8 transcript level in that cluster?

The clusters have since been updated with the new data added.

10. Line 353 to 356 “Comparisons...low milk production” seems out of context/no in line with theme of the manuscript.

We hope that our manuscript will be read by mammary gland biologists interested in lactation and breast cancer research, hence we believe this is an important point to make in our manuscript. We have updated the sentence to make it more clear. See lines: 334-336.

11. Can authors explain in the methods more on cell recovery from milk? How much milk is needed to extract that many cells? Did they use a blocking buffer during staining? Esp since lactocytes are sticky and may cause background staining. Moreover, I did not see used of viability marker for FACS. Can the authors pls provide staining with isotype controls?

More detail has been added to the “Human milk cell isolation” section. See lines 388-391. Isotype controls were not used as the FACS panel referred to has been well established for mammary cells. DAPI was used as a viability marker in the folate receptor panel (see methods section, line 425) and the negative control for the folate receptor can be seen in Supplementary Figure 7b.

12. FACS ARIA nozzle is 100um and not in nm.

✓ See line 431.

13. Line 515: there is typo in “were”.

✓ Fixed.

References

Bach K, Pensa S, Grzelak M, Hadfield J, Adams DJ, Marioni JC, Khaled WT. Differentiation dynamics of mammary epithelial cells revealed by single-cell RNA sequencing. *Nat Commun.* 2017 Dec 11;8(1):2128. doi: 10.1038/s41467-017-02001-5. PMID: 29225342; PMCID: PMC5723634.

Jin S, Guerrero-Juarez CF, Zhang L, Chang I, Ramos R, Kuan CH, Myung P, Plikus MV, Nie Q. Inference and analysis of cell-cell communication using CellChat. *Nat Commun.* 2021 Feb 17;12(1):1088. doi: 10.1038/s41467-021-21246-9.

Lim E, Vaillant F, Wu D, Forrest NC, Pal B, Hart AH, Asselin-Labat ML, Gyorki DE, Ward T, Partanen A, Feleppa F, Huschtscha LI, Thorne HJ; kConFab, Fox SB, Yan M, French JD, Brown MA, Smyth GK, Visvader JE, Lindeman GJ. Aberrant luminal progenitors as the candidate target population for basal tumor development in BRCA1 mutation carriers. *Nat Med.* 2009 Aug;15(8):907-13. doi: 10.1038/nm.2000. Epub 2009 Aug 2. PMID: 19648928.

van Amerongen R. Behind the Scenes of the Human Breast Cell Atlas Project. *J Mammary Gland Biol Neoplasia*. 2021 Mar;26(1):67-70. doi: 10.1007/s10911-021-09482-7. Epub 2021 Apr 29. PMID: 33914224; PMCID: PMC8081765.

Reviewers' Comments:

Reviewer #2:

Remarks to the Author:

The authors have not addressed my concerns. The authors have performed scRNAseq at different stages of mouse mammary gland development but not on cells isolated from the milk. This together with the analyses in at least one other specie will be required to discard putative artifactual results based on the isolation method for human cells.

Reviewer #3:

Remarks to the Author:

I thought the authors did an excellent job of answering these reviewers. The genomic data will also provide an great resource for the wider community and I hope it will not be too tedious or beaucratic for others to access.

Reviewer #4:

Remarks to the Author:

The authors state that their data will help understand parity-associated changes in the mammary gland that reduce breast cancer risk. However, the dying cells in the milk will not contribute to any tumor development. Thus, these data are not really relevant to breast cancer risk and don't even improve our understanding of mammary gland biology. They should have studied mammary epithelial progenitors/stem cells as well as other cell types that may regulate them. The revisions made to the paper addressed some of the major comments of reviewer 1, but not reviewer 2, and concerns about data quality, novelty, and biological insights remain.

Major points:

The finding that the milk contains luminal cells is expected, since these are the cells lining the ducts&alveoli where milk is produced and transported.

The breast tissue that the authors used as controls came from postmenopausal women, much older than the nulliparous breast tissue and the milk donors. Thus, they cannot be used for comparison. Several prior studies have described that age, especially menopausal status, has major impact on mammary epithelial cells. Examples include:
<https://pubmed.ncbi.nlm.nih.gov/33378681/>

The number of cells sequenced from the milk and tissues are very different when considering that the tissue is many more cell types, while the milk mostly has luminal cells. The authors have to control for depth of read/cell when comparing cell types from these two tissue types.

Cells in the milk are most likely dead/dying cells with poor RNA content. Have the authors assessed viability in a quantitative manner – not just state that some cells can grow out? In addition, the mtDNA filtering should be done more rigorously by including a plot of mtDNA reads/cell and show what cut of they used (not deviation from mean, since if all cells dead this is meaningless) and the authors should show these data, since it's critical for the assessment of data quality and reliability.

Response to reviewers' comments

Below you will find the **reviewers' comments in bold**, and our *point-by-point response italicised below*.

Reviewer #2 (Remarks to the Author):

The authors have not addressed my concerns.

We have attempted to clarify points made by the reviewer in their original feedback (quoted in italicised bold text below) as well as the new points they have raised.

“Conceptually, the use of cells present in milk makes an indirect approach to study the lactating mammary gland, however, it is unclear whether these cells recapitulate the complexity of all the different cell subtypes present in a fully functional lactating gland, preventing the comparison between both sources and questioning the conclusions extracted in this work.”

We believe the presence of live and functionally diverse cell types in human milk is of significance and should be studied even without any comparisons. Cells in the milk provide information on the secretory and immune cell types in the lactating mammary gland which has not yet been examined due to the difficulty in obtaining tissue from lactating women, something that is near impossible if not unethical. However, we do compare the cells in the milk to the non-lactating tissue as that is the resting state of this tissue, which is a relevant benchmark.

“Are these cells actually damaged cells that are being eliminated from the healthy tissue?”

As supported by data in our manuscript, we do not find evidence to suggest that the cells in the milk are damaged, and instead find that their RNA profile is of a similar high quality to cells taken directly from the breast tissue. We have provided evidence to support that the cells in milk are viable, gathered by using three different techniques:

- 1) We have performed FACS analysis on viably frozen cells from different donors, where we included the cell viability stain DAPI, to show that on average 46% of cells in the milk are live epithelial cells (see updated lines 167-169 and Supplementary Figure 9). Importantly, this fits within a similar range of post-thawed breast tissue cells, which have a range of 40-80% live cells. Of this between 5-64% cells are luminal, as determined by flow cytometry [1].*
- 2) We have isolated and cultured milk cells, as shown in Figure 1b. To further support this statement, we have provided additional data in this revised version of the manuscript (Supplementary Figure 2), which shows images of cultured cells isolated from 10 different milk cell donors and 10 different breast tissue donors. Thus, providing overwhelming evidence that milk cells can be reproducibly cultured across several biological samples.*
- 3) We applied standardised quality control techniques, routinely used across scRNA-seq analysis, to ensure that only the highest quality milk and breast tissue cells were retained for downstream analysis and interpretation. Through this analysis, we verified that cells in milk have a similar high quality viable profile to breast tissue cells, as indicated by the UMI/gene count and % mitochondrial count per cell (see new Supplementary Figure 3). If low quality cells from either milk or breast were sequenced, they were filtered out as part of our quality control analysis (see methods section). Firstly, cells from each batch of samples (containing both milk cells and breast tissue) underwent pre-processing, using the “emptyDroplets” function from the DropletUtils package in R, which removed droplets containing low counts and therefore likely ambient RNA. Following this, the cells were filtered to ensure that only high-quality cells with unique molecular identifiers (UMIs) >1000 (UMIs indicate unique mRNA reads) and percentage mitochondrial counts lower than 1x the median absolute deviation (MAD) were retained. Following filtering, the expression values for each batch were normalised and log transformed using the “computeSumFactors” from scran and “logNormCounts” from the scater package. After careful filtering, using uniform strict thresholding of cells across all samples, normalisation steps were*

undertaken for each sample in each batch and subsequently combined and normalised before downstream analysis was performed.

Taken together, we are confident in reporting that the cells in milk are viable and of a comparable high quality to cells of the normal breast tissue.

The authors have performed scRNAseq at different stages of mouse mammary gland development but not on cells isolated from the milk. This together with the analyses in at least one other specie will be required to discard putative artifactual results based on the isolation method for human cells.

Regarding comparing our data to other species, we believe that this request is outside the scope of this study. As pointed out by this reviewer, scRNA-seq data does not currently exist on cells from the mammary gland and milk cells from the same animal. Whilst this data would be interesting, to set up an experiment to acquire this data would be a significant undertaking (requiring substantial work to set up the ethics and collections for these studies) which we believe would represent an entirely separate study to what we have undertaken here.

Regarding potential artefactual results that may have arisen due to the different isolation methods (cells coming from either tissue dissociation or directly from milk), we went to great lengths to ensure that we analysed only the highest quality data and have previously addressed this point in our comments to Reviewer 1. Careful filtering and normalisation steps were performed uniformly across milk and breast tissue cell samples (see comments above for details) within each batch before combining all samples and performing batch corrections (see methods section). Intersecting genes between the batches were retained and normalisation factors for each batch were calculated. Batch effects were removed between both the sample batches and between the milk and breast tissue cells by performing mutual nearest neighbours (MNN) correction using the “fastMNN” function from the batchelor package. After these correction steps, we noted that using an MNN batch correction allowed for the immune cell subtypes to overlap between the milk and breast tissue cells, where we believe that we performed adequate correction to remove potential artefactual results arising from cells being isolated through different methods.

Reviewer #3 (Remarks to the Author):

I thought the authors did an excellent job of answering these reviewers. The genomic data will also provide an great resource for the wider community and I hope it will not be too tedious or beaucratic for others to access.

The authors thank the reviewer for recognising the value of their work and for appreciating the care the authors took in answering the reviewers’ concerns. We now include links to all raw data (uploaded to Array Express) and a link to a user-friendly website that can be used by the community to mine this rich dataset.

Reviewer #4 (Remarks to the Author):

The authors thank the reviewer for their feedback and for highlighting concepts that need to be clarified for the audience of this work.

“The authors state that their data will help understand parity-associated changes in the mammary gland that reduce breast cancer risk. However, the dying cells in the milk will not contribute to any tumor development.”

*As supported by data in our manuscript, **we do not** find evidence to suggest that the cells in the milk are dying and instead find that their RNA profile is of a similar high quality to cells taken directly from the*

breast tissue. We have provided evidence to support that the cells in milk are alive and viable, gathered by using three different techniques:

- 1) We have performed FACS analysis on viably frozen cells from different donors, including the cell viability stain DAPI, to show that on average 46% of cells in the milk are live epithelial cells (see lines 167-169 and Supplementary Figure 9). Importantly, this fits within a similar range of post-thawed breast tissue cells, which have a range of 40-80% live cells. Of this between 5-64% cells are luminal cells, as determined by flow cytometry [1].
- 2) We have isolated and cultured milk cells, as shown in Figure 1b. To further support this statement, we have provided additional data in this revised version of the manuscript (Supplementary Figure 2), which shows images of cultured cells isolated from 10 different milk cell donors and 10 different breast tissue donors. Thus, providing overwhelming evidence that milk cells can be reproducibly cultured across several biological samples.
- 3) We applied standardised quality control techniques, routinely used across scRNA-seq analysis, to ensure that only the highest quality milk and breast tissue cells were retained for downstream analysis and interpretation. Through this analysis, we verified that cells in milk have a similar high quality viable profile to breast tissue cells, as indicated by the UMI/gene count and % mitochondrial count per cell (see new Supplementary Figure 3). If low quality cells from either milk or breast were sequenced, they were filtered out as part of our quality control analysis (see methods section). Firstly, cells from each batch of samples (containing both milk cells and breast tissue) underwent pre-processing, using the “emptyDroplets” function from the DropletUtils package in R, which removed droplets containing low counts and therefore likely ambient RNA. Following this, the cells were filtered to ensure that only high-quality cells with unique molecular identifiers (UMIs) >1000 (UMIs indicate unique mRNA reads) and percentage mitochondrial counts lower than 1x the median absolute deviation (MAD) were retained. Following filtering, the expression values for each batch were normalised and log transformed using the “computeSumFactors” from scran and “logNormCounts” from the scater package. After careful filtering, using uniform strict thresholding of cells across all samples, normalisation steps were undertaken for each sample in each batch and subsequently combined and normalised before downstream analysis was performed.

Taken together, we are confident in reporting that the cells in milk are viable and of a comparable high quality to cells of the normal breast tissue. We understand that this may be a surprising fact and is indeed a paradigm shift in the field given that, despite little supporting evidence, many believed that cells in the milk are dead. We believe our findings enables a new avenue of research, which up to this point remains unappreciated within the mammary gland biology field.

“Thus, these data are not relevant to breast cancer risk and don’t even improve our understanding of mammary gland biology.”

Luminal progenitor cells have been long established as the cell of origin of basal like breast cancer [2, 3] and milk cells enable us to non-invasively collect functional human luminal cells in a non-invasive manner and perform further analysis that will impact our understanding of breast cancer in general (i.e. how do breast cancer cells compare to their functional counterpart which are lactation associated mammary cells). Our study shows that this important question can be addressed in an ex vivo manner, as it is prohibitively rare to obtain tissue samples from lactating women. In addition, pathways activated during lactation might very well be either suppressed or reactivated during breast cancer development, given their central role in the functioning of the mammary gland, and the profound tissue remodelling the entire gland undergoes during lactation. Thus, a better understanding of the molecular basis of human lactation as pioneered by our study appears fundamental for both advancing mammary gland biology and basic breast cancer research. To clarify this point, we have modified our discussion in lines 338-348 to clearly address this point.

“They should have studied mammary epithelial progenitors/stem cells as well as other cell types that may regulate them.”

We have analysed our scRNAseq data with this question in mind. As we have highlighted in our results presented in Figure 4, we have shown that **cells in milk have a similar transcriptional profile to luminal progenitor cells**. To our knowledge this is the first report to describe this feature of the epithelial cells in the milk.

“The revisions made to the paper addressed some of the major comments of reviewer 1, but not reviewer 2, and concerns about data quality, novelty, and biological insights remain.”

After taking 9 months to carefully address **all** the concerns raised by reviewer 1, recognised by reviewer 3 (see above), we were disappointed that our revisions were not personally assessed by reviewer 1. Here, we provided extra information in response to the original comments from reviewer 2 and provide feedback on concerns raised by this reviewer. Following guidance from the editor, performing experiments to compare our findings to similar studies in other species is not required and we believe is beyond the scope of this current study.

Major points:

“The finding that the milk contains luminal cells is expected, since these are the cells lining the ducts & alveoli where milk is produced and transported.”

Whilst it may be expected that milk contains luminal cells, we have provided the evidence to show for the first time, that there are in fact two secretory luminal cell populations in human milk. In addition, we also report the identity of **all the other** cell types found in the milk, in particular the immune cells. We also perform extensive analysis on how the epithelial cells in the milk are regulated and how they potentially interact with the immune cells in the milk. All of which is novel and has never been previously reported.

“The breast tissue that the authors used as controls came from postmenopausal women, much older than the nulliparous breast tissue and the milk donors. Thus, they cannot be used for comparison. Several prior studies have described that age, especially menopausal status, has major impact on mammary epithelial cells. Examples include: <https://pubmed.ncbi.nlm.nih.gov/33378681/>”

Within our study we have provided scRNA-sequencing on breast tissue cells taken from women aged 19-65 years old, compared to cells from milk taken from women aged 27-44 years old (Supplementary Figure 4a). The reviewer is correct in saying that breast tissue samples taken from parous women were older. However as provided in our response to reviewer 3, we did not find significant differences in the gene expression profile of luminal progenitor cells from parous compared to nulliparous women (see Reviewer Figure 3a). Nor did we identify differences between uniparous and nulliparous samples in dimensional reduction plots such as UMAPs (Supplementary Figure 6a) or principal component analysis (Supplementary Figure 4b). It should be noted that the referenced study, whilst being excellent, has been conducted on mice and similar data has not been published for human samples.

“The number of cells sequenced from the milk and tissues are very different when considering that the tissue is many more cell types, while the milk mostly has luminal cells. The authors have to control for depth of read/cell when comparing cell types from these two tissue types.”

As the reviewer has stated, we did indeed find that the proportions of cell types found in the milk were different to those found in the breast. However, the authors did not set out to compare proportions of the cell subtypes between milk and breast, but rather report the data as is (see Supplementary Figure 6c). Through our extensive experience working with mammary scRNA-seq data, we have found that the proportion of different cell types has little bearing on the overall depth or read/cell, as this is more significantly impacted by the proportions in which the barcoded RNA from different samples is mixed and overall sequencing depth. To control for differences in numbers of cells or sequencing depth per cell, which arises as part of any scRNA-seq study, we have taken care to perform the same quality control and normalisation steps on samples from breast and milk using standard analysis techniques (see methods). Briefly, cells from each batch of samples (containing both milk cells and breast tissue) underwent pre-

processing, using the “emptyDroplets” function from the DropletUtils package in R, which removed droplets containing low counts and therefore likely ambient RNA. Following this, the cells were filtered to ensure that only high-quality cells with unique molecular identifiers (UMIs) >1000 (UMIs indicate unique mRNA reads) and percentage mitochondrial counts lower than 1x the median absolute deviation (MAD) were retained. Following filtering, the expression values for each batch were normalised and log transformed using the “computeSumFactors” from scran and “logNormCounts” from the scater package. After careful filtering, using uniform strict thresholding of cells across all samples, normalisation steps were undertaken for each sample in each batch and subsequently combined and normalised before downstream analysis was performed.

“Cells in the milk are most likely dead/dying cells with poor RNA content. Have the authors assessed viability in a quantitative manner – not just state that some cells can grow out?”

As commented above, throughout our extensive analysis, we find no evidence to suggest that the majority of cells in milk are dead or dying. Together with providing additional data that milk cells can grow out across a multitude of milk samples from different women (Figure 1b and Supplementary Figure 2), we have provided FACS data showing that an abundance of epithelial cells are alive (Supplementary Figure 9b) and carefully performed quality control, filtering and normalisation to ensure that we only conducted downstream analysis on high quality (high UMI/gene counts per cell) and viable (low % mitochondrial reads per cell) cells from milk and breast tissue (see updated data on pre- and post-filtering/normalisation of cells in Supplementary Figure 3).

“In addition, the mtDNA filtering should be done more rigorously by including a plot of mtDNA reads/cell and show what cut of they used (not deviation from mean, since if all cells dead this is meaningless) and the authors should show these data, since it’s critical for the assessment of data quality and reliability.”

Upon receiving this comment, we updated our Supplementary Figures to provide additional data on the UMI, gene and % mitochondrial count per cell (coloured by sample) pre- and post- filtering/normalisation of our data (see Supplementary Figure 3). As mentioned in the methods, we performed filtering on the cells per batch (thresholds on the pre-filtering/normalisation plots now displayed in Supplementary Figure 3), where we only retained cells for downstream analysis that had high UMI counts (>1000 UMIs) and less than 1 MAD % mitochondrial RNA per cell. This resulted in 54,714 and 56,030 post-filtered, high quality NMCs and LMCs, respectively (Figure 1b) that were further interrogated as part of this study and reported in this manuscript. It may be interesting for the reviewer to note that the majority of the post-filtered cells with the higher mitochondrial counts are actually breast tissue cells (blue) and not milk cells (pink) (Supplementary Figure 3cii-v).

References

1. Engelbrecht, L.K., et al., A strategy to address dissociation-induced compositional and transcriptional bias for single-cell analysis of the human mammary gland, in *bioRxiv*. 2020.
2. Molyneux, G., et al., BRCA1 basal-like breast cancers originate from luminal epithelial progenitors and not from basal stem cells. *Cell Stem Cell*, 2010. **7**(3): p. 403-17.
3. Lim, E., et al., Aberrant luminal progenitors as the candidate target population for basal tumor development in BRCA1 mutation carriers. *Nat Med*, 2009. **15**(8): p. 907-13.

Reviewers' Comments:

Reviewer #4:

Remarks to the Author:

The authors have responded to each of the reviewers' specific comments and included additional data analyses to control for quality of the data, batch effects, etc. These clearly show sample-to-sample technical variabilities, which the authors addressed by computational tools. However, they cannot change the fact that the samples are coming from women with different age and parity. The fact that they do not see parity-related differences raises concerns and their statement that "). It should be noted that the referenced study, whilst being excellent, has been conducted on mice and similar data has not been published for human samples." is not correct. Parity-related differences in expression profiles have been described by many groups in several prior papers using different methods even pre-RNA-seq era.

At the end the manuscript describes scRNAseq data of normal human breast tissue and milk (both of which has already been done and published) from a heterogeneous cohort, w/o any additional experiments, and w/o any new biological insights.